# PLUG-AND-PLAY POLICY PLANNER FOR LARGE LANGUAGE MODEL POWERED DIALOGUE AGENTS

**Yang Deng**[1] **Wenxuan Zhang**[*] **Wai Lam**[2] **See-Kiong Ng**[1] **Tat-Seng Chua**[1]
[1]National University of Singapore [2]The Chinese University of Hong Kong
`{ydeng, seekiong, dcscts}@nus.edu.sg`
`isakzhang@gmail.com wlam@se.cuhk.edu.hk`

## ABSTRACT

Proactive dialogues serve as a practical yet challenging dialogue problem in the era of large language models (LLMs), where the dialogue policy planning is the key to improving the proactivity of LLMs. Most existing studies enable the dialogue policy planning of LLMs using various prompting schemes or iteratively enhance this capability in handling the given case with verbal AI feedback. However, these approaches are either bounded by the policy planning capability of the frozen LLMs or hard to be transferred to new cases. In this work, we introduce a new dialogue policy planning paradigm to strategize LLMs for proactive dialogue problems with a tunable language model plug-in as a plug-and-play dialogue policy planner, named PPDPP. Specifically, we develop a novel training framework to facilitate supervised fine-tuning over available human-annotated data as well as reinforcement learning from goal-oriented AI feedback with dynamic interaction data collected by the LLM-based self-play simulation. In this manner, the LLM-powered dialogue agent can not only be generalized to different cases after the training, but also be applicable to different applications by just substituting the learned plug-in. In addition, we propose to evaluate the policy planning capability of dialogue systems under the interactive setting. Experimental results demonstrate that PPDPP consistently and substantially outperforms existing approaches on three different proactive dialogue applications, including negotiation, emotional support, and tutoring dialogues.[1]

## 1 INTRODUCTION

Large language models (LLMs) powered dialogue agents (*e.g.*, ChatGPT (Ouyang et al., 2022), Vicuna (Chiang et al., 2023), LLaMA2-Chat (Touvron et al., 2023), etc) have demonstrated exceptional proficiency in context understanding and response generation in various dialogue problems (Bang et al., 2023; Zhang et al., 2023b; Zhao et al., 2023). However, as LLMs are trained to passively follow users' instructions, dialogue agents built upon them typically prioritize accommodating users' intention. Therefore, LLM-powered dialogue agents often face challenges in handling proactive dialogue problems that require the dialogue agent to strategically take the initiative to steer the conversation towards an anticipated goal (Deng et al., 2023a), such as negotiation (Zhan et al., 2022), emotional support (Liu et al., 2021; Zheng et al., 2023), and tutoring (Macina et al., 2023).

In such scenarios, the key is to improve the capability of LLM-powered dialogue agents in dialogue policy planning, which refers to the process of deciding what actions the dialogue agent should take to effectively achieve specific goals during the dynamic interactions with the user. In the pre-LLM era, researchers mainly employ corpus-based learning approaches to conduct the dialogue policy planning via dialogue act prediction (Zhou et al., 2020; Joshi et al., 2021; Cheng et al., 2022; Wang et al., 2023c; Deng et al., 2023c; 2022). However, such approaches rely heavily on static human-annotated dialogues and fail to optimize the long-term goal of the conversation. With the advent of LLMs, it further becomes unrealistic and costly to fine-tune the whole dialogue systems for every specific application. To this end, as shown in Figure 1(a), recent works investigate prompt-based

---

[*]Corresponding author.
[1]The code can be accessed via `https://github.com/dengyang17/PPDPP`.

policy planning methods that prompt a frozen actor LLM to either conduct self-thinking of strategy planning for each turn (Zhang et al., 2023a; Deng et al., 2023b; Wang et al., 2023a) or generate AI feedback given the whole dialogue history to iteratively improve the dialogue policy planning for a certain case (Fu et al., 2023; Yu et al., 2023).

Despite their effectiveness in improving the dialogue policy planning, there are several challenges that remain to be tackled. 1) LLMs fall short of planning effective dialogue policy with zero-shot or few-shot prompting schemes (Deng et al., 2023b). Therefore, the improvement of accomplishing the goal will be limited by the planning capability of the frozen actor LLM. 2) Existing approaches based on iterative refinement (Fu et al., 2023; Yu et al., 2023) lack of transferability, as multiple rounds of self-play dialogue simulations are required for every new-coming case to plan a satisfactory strategy for it, which is impractical in real-world applications. 3) Existing studies typically evaluate the performance of dialogue agents in terms of turn-level response quality measurements based on fixed reference responses. However, these evaluation protocols fail to automatically assess the policy planning capability of the dialogue agent, which is determined by the effectiveness and efficiency of the goal achievement in multi-turn conversations.

To tackle these challenges, we introduce a novel dialogue policy planning paradigm to strategize LLMs with a tunable language model plug-in, named Plug-and-Play Dialogue Policy Planner (PPDPP). As shown in Figure 1(b), PPDPP acts as the policy agent to predict the dialogue strategy at the next turn for the dialogue agent, which can be first supervisedly fine-tuned with available human-annotated corpora. Then, we employ the self-play paradigm to prompt two LLMs (an assistant and a user) with various case background information to perform the role-playing conversation that simulates the dynamic environment of multi-turn interactions between the dialogue agent and the real user. For each case, these two LLMs are each tasked with distinct, often competing goals (*e.g.*, in negotiation dialogues, the buyer seeks to attain a more favorable price, whereas the seller endeavors to secure a higher price). Meanwhile, a third LLM acts as the reward model to provide goal-oriented verbal feedback, indicating the goal achievement, which is transformed to scalar rewards used for reinforcement learning (RL). When reaching the goal or the maximum conversation turn, we leverage RL algorithm to further tune the policy agent with the collected interaction data and the goal-oriented AI feedback. In this way, the LLM-powered dialogue agent can not only exhibit more adaptability to various new cases than prompt-based approaches, but also find utility across diverse applications simply by shifting the tuned plug-in without affecting the LLM's exceptional capabilities of context understanding and response generation.

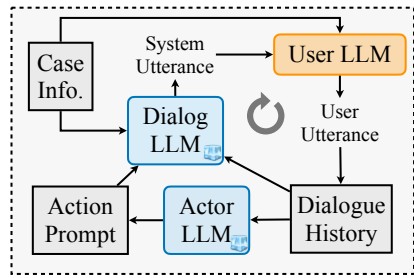

(a) Prompt-based Policy Planning

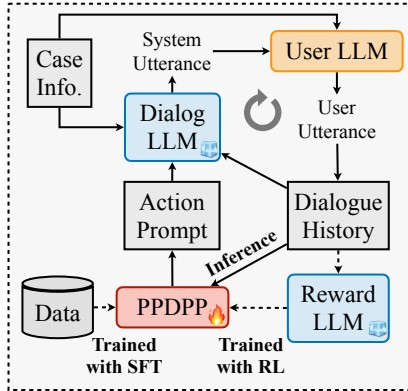

(b) Plug-and-Play Dialogue Policy Planner

Figure 1: The architectures of two types of LLM-based proactive dialogue systems. Dashed lines will be blocked during the inference phase.

To overcome the limitation of traditional turn-level response evaluation metrics, we further propose an LLM-based interactive evaluation approach that harnesses LLM-based user simulators and reward models as introduced above. This approach enables the simulation of diverse user-assistant interactions to assess both the success rate and the average number of turns of achieving designated goals. We conduct extensive experiments on three different proactive dialogue problems, including negotiation, emotional support, and tutoring dialogues. Experimental results demonstrate the superiority of the proposed PPDPP framework over existing LLM-based dialogue systems, showing that PPDPP can effectively and efficiently lead the conversations to achieve the designated goal.

Table 1: Overview of LLM-based general (upper) and dialogue (lower) policy planning methods.

| | Subjective Goal | Multi-turn Interaction | Decision Making | Explicit Strategy | Long-term Optimization | Gradient Learning | Transfer-ability |
|---|---|---|---|---|---|---|---|
| DSP (Li et al., 2023) | ✗ | ✗ | ✗ | ✗ | ✓ | ✓ | ✓ |
| RAP (Hao et al., 2023) | ✗ | ✓ | ✓ | ✓ | ✓ | ✗ | ✗ |
| Reflexion (Shinn et al., 2023) | ✗ | ✗ | ✓ | ✗ | ✓ | ✗ | ✗ |
| Retroformer (Yao et al., 2023) | ✗ | ✗ | ✓ | ✗ | ✓ | ✓ | ✓ |
| MI-Prompt (Chen et al., 2023) | ✓ | ✗ | ✗ | ✓ | ✗ | ✗ | ✗ |
| Ask-an-Expert (Zhang et al., 2023a) | ✓ | ✗ | ✓ | ✗ | ✗ | ✗ | ✓ |
| ProCoT (Deng et al., 2023b) | ✓ | ✗ | ✓ | ✓ | ✗ | ✗ | ✓ |
| ICL-AIF (Fu et al., 2023) | ✓ | ✓ | ✗ | ✗ | ✓ | ✗ | ✗ |
| **PPDPP** | ✓ | ✓ | ✓ | ✓ | ✓ | ✓ | ✓ |

## 2 RELATED WORKS

**Dialogue Policy Planning**   Dialogue policy planning has been widely-studied in task-oriented dialogues (Jang et al., 2022; Feng et al., 2023) and conversational recommendation (Gao et al., 2021; Deng et al., 2021), where the interaction process can be easily abstracted into a sequence of slots and values (*e.g.*, location, price, etc). Meanwhile, the success of planning is objective, such as whether the system provides an appropriate entity/item. However, in proactive dialogues (Deng et al., 2023a; Liao et al., 2023), there is no pre-defined agenda or schema for simplifying the multi-turn interaction. Instead, the natural language interaction requires more complex reasoning and certain domain knowledge (*e.g.*, psychological or pedagogical skills). Moreover, the planning outcome is rather subjective, such as learning gain during tutoring, emotional intensity relaxation during counselling. Therefore, it imposes more difficulties in planning optimal dialogue policy in proactive dialogues. In order to mimic the behaviors of human experts, corpus-based fine-tuning approaches are typically adopted for predicting the dialogue strategies (Joshi et al., 2021; Cheng et al., 2022; Wang et al., 2023c). As summarized in Table 1, we differentiate our method from the recent LLM-based policy planning methods in terms of seven perspectives. General policy planning methods typically optimize towards an objective goal in a single-turn interaction, such as ROUGE score in summarization (Li et al., 2023) or accuracy in QA (Shinn et al., 2023; Yao et al., 2023). As for dialogue policy planning methods, Chen et al. (2023) validate the effectiveness of mixed-initiative strategy-based prompting in proactive dialogue problems. Some methods Wang et al. (2023b); Deng et al. (2023b); Zhang et al. (2023a) prompt LLMs to conduct self-thinking of policy planning for the next turn, ignoring the long-term conversation goals. Fu et al. (2023) conduct self-play simulation to iteratively refine the policy planning with long-term feedback. However, this type of iterative refinement is exclusive to each individual case, but not transferable to new situations. Moreover, the policy planning capability in LLM-powered dialogue agents cannot be improved by these methods, as all parameters are frozen and not learnable. As for the proposed PPDPP, a learnable language model plug-in can be fine-tuned for improving the policy planning capability without affecting other functionalities of LLM-powered dialogue agents.

**Learnable Plug-ins for Large Language Models**   Due to the black-box nature of commercial LLMs and the high expenses of fine-tuning the whole open-source LLMs, a recent trend in improving certain capabilities of LLMs is to investigate the utility of external plug-ins, such as APIs (Schick et al., 2023), vision models (Wu et al., 2023), or functional models from Huggingface (Shen et al., 2023). However, these plug-ins fail to learn from valuable feedback to iteratively enhance the their capabilities, resulting in performances that are solely dependent on the quality of fixed plug-ins. Some latest studies demonstrate that small language models are also valuable plug-ins for LLMs in various applications, such as text classification (Xu et al., 2023), summarization (Li et al., 2023), question answering (Yao et al., 2023), or to endow LLMs with specific capabilities, such as mental state reasoning (Sclar et al., 2023). These learnable plug-ins can benefit from supervised fine-tuning with available resources or reinforcement learning with the environment.

**Reinforcement Learning from AI Feedback**   As LLMs become more powerful to be capable of supervising other models, Bai et al. (2022) propose the idea of "RL from AI Feedback" (RLAIF) to train a harmless and detoxified LLM through self-improvement, without human labels. Since the feedback from LLMs is typically in the form of natural language, most of existing studies (Shinn

et al., 2023; Fu et al., 2023; Madaan et al., 2023; Hao et al., 2023) directly leverage the generated natural language feedback from LLMs to self-refine the task instruction prompt, instead of obtaining a scalar reward for training the model. In this work, we propose the goal-oriented AI feedback for facilitating RLAIF under the context of dialogue systems, which not only transforms the textual feedback into scalar rewards, but also capture long-term goal-oriented rewards that obtain from the dynamic multi-turn interactions, instead of AI preference on single-turn responses.

## 3 METHOD

**MDP Environment**  We formulate the dialogue process as a Markov Decision Process (MDP). At each turns $t$, according to the observation on the dialogue history, the dialogue system selects an action $a_t \in \mathcal{A}$, where $\mathcal{A}$ is a set of candidate strategies pre-defined by domain experts. In return, the user player responds to the action. This process repeats until the conversational goal is achieved or the maximum number of turns $T$ is reached. The objective is to learn a policy $\pi$ maximizing the expected cumulative rewards over the observed dialogue episodes as:

$$\pi^* = \arg\max_{\pi \in \Pi} \left[ \sum_{t=0}^{T} r(s_t, a_t) \right], \tag{1}$$

where $s_t$ is the state representing the dialogue history. $r(\cdot)$ is the intermediate reward, denoted as $r_t$.

**Plug-and-Play Dialogue Policy Planner**  As shown in Figure 1(b), a smaller model is adopted to act as a plug-in for controlling the dialogue policy planning in the LLM-powered dialogue agent. We leverage a tunable pre-trained language model, *e.g.*, RoBERTa (Liu et al., 2019), as the dialogue policy planner to predict the action $a_t$. Before performing interactive online learning, PPDPP can be initialized by **supervised fine-tuning** (SFT) on available dialogue corpus $\mathcal{D}$. Specifically, given the dialogue history $\{u_1^{\text{sys}}, u_1^{\text{usr}}, ..., u_{t-1}^{\text{sys}}, u_{t-1}^{\text{usr}}\}$ as the current state $s_t$, the SFT process aims to minimize the cross entropy loss between the predicted action $a_t$ and the human-labeled action $y_t$ for each turn $t$ in the annotated dialogue:

$$a_t = \textbf{PPDPP}(u_1^{\text{sys}}, u_1^{\text{usr}}, ..., u_{t-1}^{\text{sys}}, u_{t-1}^{\text{usr}}) \tag{2}$$

$$\mathcal{L}_c = -\frac{1}{|\mathcal{D}|} \sum_{d \in \mathcal{D}} \frac{1}{T_d} \sum_{t=1}^{T_d} a_t \log y_t \tag{3}$$

where $T_d$ denotes the number of turns of the dialogue. Although corpus-based learning typically leads to sub-optimal policy, such initialization is supposed to accelerate the convergence process of interactive online training.

**Self-play Interaction**  During the interactive online learning, we prompt two LLMs as the user and the assistant to perform self-play conversations that simulate the dynamic user-assistant interaction. The descriptions of the role and the instructions of their corresponding conversational goals are delivered to each LLM. For example, in emotional support dialogues, the patient (user) player will receive the situation description about the causes of the emotional problem while the therapist (assistant) player will receive the task description to reduce users' emotional distress and help them work through the challenges. In tutoring dialogues, the student (user) player will receive the descriptions about their knowledge state, while the teacher (assistant) player will receive the task descriptions to teach users to master a certain exercise. When it comes to the assistant's turn, PPDPP first predict the next action $a_t$ based on the interaction history. The predicted action is mapped to a pre-defined natural language instruction $\mathcal{M}_a(a_t)$. Then the assistant player generates the strategic response based on the dialogue history and the natural language action instruction:

$$u_t^{sys} = \textbf{LLM}_{\text{sys}}(p_{\text{sys}}; \mathcal{M}_a(a_t); u_1^{\text{sys}}, u_1^{\text{usr}}, ..., u_{t-1}^{\text{sys}}, u_{t-1}^{\text{usr}}) \tag{4}$$

Then the user player generates the response based on the updated dialogue history with $u_t^{sys}$:

$$u_t^{usr} = \textbf{LLM}_{\text{usr}}(p_{\text{usr}}; u_1^{\text{sys}}, u_1^{\text{usr}}, ..., u_{t-1}^{\text{sys}}, u_{t-1}^{\text{usr}}, u_t^{sys}) \tag{5}$$

where $p_{\text{sys}}$ and $p_{\text{usr}}$ are the corresponding prompts. This process is repeated until a terminal state is reached. Overall, there are three types of states in the self-play interaction: 1) ON-GOING: the dialogue between the two players is still ongoing as the goal has not been achieved; 2) GOAL-COMPLETED: the designated conversational goal is completed, such as solving the seeker's emotional problem or the student mastering the exercise; 3) GOAL-FAILED: the conversational goal is considered as failure when the conversation reaches a maximum turn without completing the goal.

**LLM as Reward Model**    We prompt a third LLM to be the reward model, named $\textbf{LLM}_{\text{rwd}}$, which has two functions: (1) to determine the goal completion during the conversation; (2) to evaluate the policy outcome with scalar rewards. Specifically, we prompt the reward model to answer a multi-choice question to generate the goal-oriented AI feedback. We further define a mapping $\mathcal{M}_r(\cdot)$ to transform verbal feedback to scalar rewards.

Due to the subjectivity of the planning outcome as well as the variance of the LLM-generated output, we follow a common practice (Wang et al., 2023e) to alleviate these issues by sampling the decoded sequences of the reward LLM. In general, we obtain a scalar value $v_t$ by sampling the goal-oriented AI feedback for $l$ times and converting them into a scalar value through averaging:

$$v_t = \frac{1}{l} \sum\nolimits_{i=1}^{l} \mathcal{M}_r(\textbf{LLM}_{\text{rwd}}(p_{\text{rwd}}; u_1^{\text{sys}}, u_1^{\text{usr}}, ..., u_{t-1}^{\text{sys}}, u_{t-1}^{\text{usr}}, u_t^{sys}, u_t^{usr}; \tau)) \tag{6}$$

where $p_{\text{rwd}}$ is the prompt. We first use $v_t$ to determine the state of the self-play interaction. If $v_t$ is not less than a certain threshold $\epsilon$, we regard the state as GOAL-COMPLETED. If the conversation reach a terminal state, including GOAL-COMPLETED and GOAL-FAILED, we have the reward $r_t = v_t$. If not, we assign a small negative reward, $e.g.$, $r_t = -0.1$, to penalize the lengthy conversation for promoting efficient goal completion.

**Reinforcement Learning**    When reaching the goal or the maximum conversation turn, we obtain goal-oriented reward $r_t$. We denote the policy agent as $\pi(a_t|s_t)$, which returns the probability of taking action $a_t$ given the state $s_t$. To optimize the policy agent, we simply use the vanilla policy gradient method (Sutton et al., 1999), formulated as follows:

$$\theta \leftarrow \theta - \alpha \nabla \log \pi_\theta(a_t|s_t) R_t \tag{7}$$

where $\theta$ denotes the parameter of the policy network, $\alpha$ denotes the learning rate of the policy network, and $R_t$ is the total reward accumulating from turn $t$ to the final turn $T$: $R_t = \sum_{t'=t}^{T} \gamma^{T-t'} r_{t'}$, where $\gamma$ is a discount factor which discounts future rewards over immediate reward.

During the inference, the tuned PPDPP directly provides the action prompt, based on the dialogue history, for guiding the dialogue LLM to generate the next response, while the reward LLM will not be used as shown in Figure 1(b). In this manner, the LLM-powered dialogue agent with the tuned PPDPP can be directly applied to diverse new situations without the necessity of performing multiple iterations of simulation for every new cases.

## 4    EXPERIMENTAL SETUPS

### 4.1    DATASETS

We evaluate the proposed framework in three different applications of proactive dialogues, including negotiation dialogues, emotional support dialogues, and tutoring dialogues. The statistics of adopted datasets are presented in Table 2. In specific, the human-annotated dialogues in the train set are used for the supervised fine-tuning of the dialogue policy planner, while only the case background information in the dataset is adopted for the reinforcement learning process.

**CraisglistBargain** (He et al., 2018) is created under the bargain negotiation setting where the buyer and the seller are negotiating the price of an item on sale. Yang et al. (2021) design 15 dialogue acts for labeling this dataset, including 11 negotiation strategies and 4 terminal acts. In our experiment, we only consider the 11 negotiation strategies and split the development set

Table 2: The statistics of datasets (train/dev/test).

| Dataset | # Case | # Act |
|---|---|---|
| CraisglistBargain | 3,090/188/188 | 11 |
| ESConv | 1,040/130/130 | 8 |
| CIMA | 909/113/113 | 5 |

into two parts as a new development set and a test set. Each case is associated with an item category, an item description, a buyer target price, and a seller target price, which are adopted as the instruction information.

**ESConv** (Liu et al., 2021) is an emotional support conversation dataset, contains 1,300 cases with 8 types of support strategies. We adopt the original train/dev/test split. Each case is accompanied with a problem type, an emotion type, and a situation description.

**CIMA** (Stasaski et al., 2020) is a crowd-sourced dataset, where annotators were asked to role-play students and teachers by working through an exercise with 5 pedagogical strategies on translating a prepositional sentence from English to Italian. We regard each exercise as a case in our experiment and randomly split the dataset into train/dev/test sets by 8:1:1.

## 4.2 EVALUATION PROTOCOLS

Previous studies (Joshi et al., 2021; Cheng et al., 2022; Wang et al., 2023c) on emotional support, negotiation, and tutoring dialogues typically evaluate the turn-level performance, based on the fixed reference responses. Differently, when it comes to the evaluation of dialogue policy planning, it would be more appropriate to focus on the dialogue-level interactive evaluation (Kwan et al., 2023; Gao et al., 2021). Goal completion is the key to evaluating proactive dialogue systems (Deng et al., 2023a). To this end, we adopt the average turn (AT) and the success rate at turn $t$ (SR@$t$) as the automatic evaluation metrics. AT measures the efficiency of the goal completion by calculating the average number of turns to achieve the goal while SR measures the effectiveness of the goal completion by computing the success rate of achieving the goal within a pre-defined maximum turn. We set the maximum turn of the conversation as 8 in our experiment. In particular, as for CraisglistBargain, since reaching a deal cannot be regarded as a success, we adopt the Sale-to-List Ratio (SL%) (Zhou et al., 2019) as the metrics to determine the effectiveness of goal completion during the negotiation dialogue, which is formulated as (deal price − seller target price)/(buyer target price − seller target price). A higher SL% represents the buyer gets more benefits from the deal. If failing to reach a deal at the end, we assign SL% as 0 for this case. The evaluation is conducted by interacting with the LLM-based user simulator, while the goal completion (*i.e.*, GOAL-COMPLETED or GOAL-FAILED) is determined by the LLM-based reward model, as introduced in Section 3. To validate the reliability of this evaluation framework, a preliminary analysis of adopting LLMs as reward models and user simulators is presented in Appendix A.

## 4.3 IMPLEMENTATION DETAILS AND BASELINES

We adopt RoBERTa (roberta-large) as the default plug-and-play dialogue policy planner for evaluation. The details of training process are provided in Appendix B. In the main results of evaluating the dialogue policy planning methods, we use ChatGPT (gpt-3.5-turbo-0613) as the frozen LLM for both the role-playing LLMs ($\mathbf{LLM}_{\text{sys}}$ and $\mathbf{LLM}_{\text{usr}}$) and the reward model ($\mathbf{LLM}_{\text{rwd}}$). As for the role-playing LLMs, we set the temperature $\tau = 0$ to generate the deterministic outputs with the same inputs. While we set $\tau = 1.1$ and the sample times $l = 10$ for the reward model to integrate the scalar rewards[2]. In order to compare with different LLMs, we also adopt two popular open-source LLM-based dialogue systems, including Vicuna-13B-delta-v1.1 and LLaMA-2-13B-Chat with the same hyper-parameter setting as ChatGPT.

The role-playing prompts for the assistant and the user players ($p_{\text{sys}}$ and $p_{\text{usr}}$) are presented in Appendix E.1 and E.2, respectively. The mapping of natural language instructions, $\mathcal{M}_a(\cdot)$, for dialogue actions is presented in Appendix E.3. The prompts for the reward model, $p_{\text{rwd}}$, and the whole reward mapping process are presented in Appendix E.4.

As for the baselines, we first compare to a general fine-tuned dialogue models, DialoGPT (Zhang et al., 2020). Furthermore, we adopt several latest LLM-based dialogue policy planning methods for comparisons, including vanilla LLM (Standard), Proactive (Deng et al., 2023b), ProCoT (Deng et al., 2023b), Ask-an-Expert (AnE) (Zhang et al., 2023a), and ICL-AIF (Fu et al., 2023). Details for these baselines are presented in Appendix E.5.

## 5 EXPERIMENTS

### 5.1 OVERALL EVALUATION

Table 3 summarizes the experimental results of method comparisons on three datasets. Overall, the proposed method, PPDPP, consistently outperforms all the baselines with a noticeable margin across

---

[2]We conduct the ablation study on the sampling strategy in Appendix D.

Table 3: Experimental results. #Tokens denotes the approximate tokens required for prompting LLMs to simulate a new conversation, where $L$ denotes the standard prompt length for one episode of conversation. $M$ and $N$ are respectively set to 3 and 5 by default as set by corresponding works.

| Method | #Tokens | CraisglistBargain | | | ESConv | | CIMA | |
| --- | --- | --- | --- | --- | --- | --- | --- | --- |
| | | AT↓ | SR↑ | SL%↑ | AT↓ | SR↑ | AT↓ | SR↑ |
| DialoGPT | - | 6.73 | 0.3245 | 0.2012 | 5.31 | 0.7538 | 5.43 | 0.4956 |
| Standard | $\mathcal{O}(L)$ | 6.47 | 0.3830 | 0.1588 | 5.10 | 0.7692 | 3.89 | 0.6903 |
| AnE (Zhang et al., 2023a) | $\mathcal{O}((M+1)L)$ | 5.91 | 0.4521 | 0.2608 | 4.76 | 0.8000 | 3.86 | 0.6549 |
| Proactive (Deng et al., 2023b) | $\mathcal{O}(L)$ | 5.80 | 0.5638 | 0.2489 | 5.08 | 0.7538 | 4.84 | 0.5310 |
| + MI-Prompt (Chen et al., 2023) | $\mathcal{O}(2L)$ | 5.74 | 0.5691 | 0.2680 | 4.78 | 0.7846 | 4.70 | 0.5664 |
| ProCoT (Deng et al., 2023b) | $\mathcal{O}(L)$ | 6.22 | 0.5319 | 0.2486 | 4.75 | 0.7923 | 4.58 | 0.5487 |
| + MI-Prompt (Chen et al., 2023) | $\mathcal{O}(2L)$ | 6.12 | 0.5532 | 0.3059 | 4.83 | 0.7769 | 4.72 | 0.5221 |
| ICL-AIF (Fu et al., 2023) | $\mathcal{O}((N+1)L)$ | 6.53 | 0.3617 | 0.1881 | 4.69 | 0.8079 | 4.19 | 0.6106 |
| **PPDPP** | $\mathcal{O}(L)$ | 5.62 | 0.6117 | **0.3376** | **4.56** | **0.8462** | **3.03** | **0.8407** |
| - w/o SFT | $\mathcal{O}(L)$ | 5.71 | 0.6223 | 0.3354 | 4.68 | 0.8384 | 3.18 | 0.8230 |
| - w/o RL | $\mathcal{O}(L)$ | **5.57** | **0.6649** | 0.2280 | 5.24 | 0.7308 | 3.41 | 0.7965 |

three different proactive dialogue problems. The results show that PPDPP can not only efficiently achieve the conversational goal (less Average Turn), but also effectively accomplish more tasks (higher Success Rate or higher Sale-to-List Ratio). Besides, compared with AnE and ICL-AIF, PPDPP requires much fewer number of tokens for calling the API of black-box LLMs to handle each new conversation. Furthermore, the supervised fine-tuning on human-annotated dialogues contribute little to the final performance, as compared to reinforcement learning with the dynamic environment. In the following, we discuss the task-specific observations for each application:

**Negotiation Dialogues (CraisglistBargain)** Among the baselines, all three turn-level policy planning methods (*incl.* AnE, Proactive, and ProCoT) substantially improve the deal success rate (SR) and the deal benefit (SL%). Unexpectedly, ICL-AIF only casts a trivial improvement on the deal benefit while even negatively affecting the deal success rate. This result indicates that the dialogue-level AI feedback fails to dynamically adjust the dialogue strategy along with the dialogue state transition to reach the consensus with the users in negotiation dialogues. As for the proposed method, we observe that PPDPP (- w/o RL) can largely improve the deal success rate with supervised fine-tuning on the human-annotated corpus, but bring not much benefit to the system side. RL with simulated interactions further increases the SL% with a substantial margin ($0.2280 \rightarrow 0.3376$) by optimizing the policy towards a higher negotiation outcome. However, this optimization objective negatively affects the deal success rate ($0.6649 \rightarrow 0.6117$), which is as expected because it inevitably downgrades the deal willingness of the seller when you are trying to maximize your own benefit.

**Emotional Support Dialogues (ESConv)** ChatGPT with Standard prompting has already achieved a quite good performance in this problem, *e.g.*, reaching a success rate of 76.92%. Among the baselines, those methods of learning from AI feedback (*incl.* AnE and ICL-AIF) perform slightly better than explicit strategy prediction methods (*incl.* Proactive and ProCoT). Compared with these zero-shot approaches, simply supervised fine-tuning on human-annotated corpus (PPDPP - w/o RL) performs even worse than Standard prompting. This result shows that corpus-based learning is far from satisfactory for handling emotional support dialogues. After learning from dynamic interactions, PPDPP effectively improves the success rate ($0.7308 \rightarrow 0.8462$) as well as outperforms all other baselines with a noticeable margin.

**Tutoring Dialogues (CIMA)** All the baseline methods fail to defeat the Standard prompting scheme in tutoring dialogues, indicating that the ChatGPT itself has already been powerful at teaching others in translating English into Italian. However, different from emotional support dialogues, the corpus-based learning is shown to be useful in tutoring dialogues, substantially outperforming all the baselines. We attribute this difference to the diversity of cases in these two applications: In ESConv, there are various emotional issues that require different chains of strategies to handle. While in CIMA, all the cases are concerned on the same type of exercises, *i.e.*, translating an English sentence with the same grammatical structure into Italian. Corpus-based learning tends to perform well in those testing cases that are similar to the training data. Nevertheless, RL can still further improve the performance to a great extent ($0.7965 \rightarrow 0.8407$).

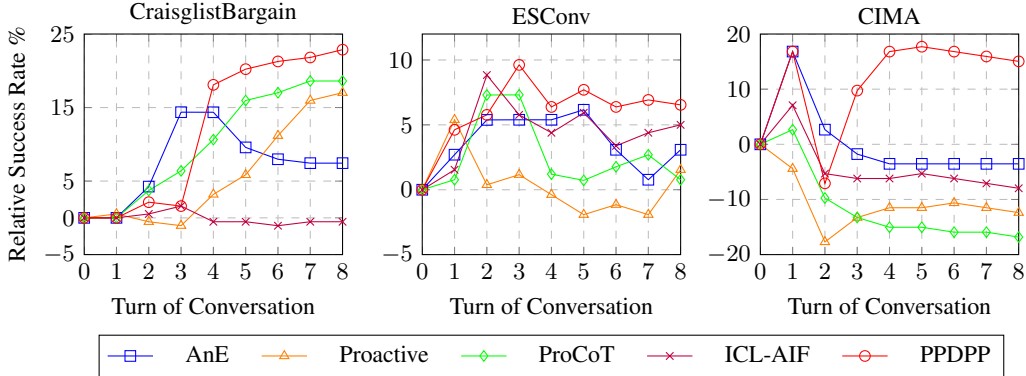

Figure 2: Comparisons of relative success rate against Standard at different conversation turns. The relative success rate is calculated by subtracting the actual success rate of the Standard prompting method from that of the concerned method.

## 5.2 IN-DEPTH ANALYSIS

**Performance w.r.t Turns** Besides SR@8, we also evaluate the success rate at each turn (SR@$t$) in Figure 2. To better compare different methods, we report the relative success rate compared with the Standard prompting schemes. For example, the line of $y = 0$ represents the curve of SR@$t$ for Standard against itself. As for the negotiation dialogues (CraisglistBargain), we also present the relative Sale-to-List Ratio against Standard in terms of different conversation turns (SL%@$t$) in Figure 3. PPDPP outperforms these baselines across all the datasets and almost each turn in the conversation session. The exception is that AnE has a relatively strong performance at the first few turns. By obtaining the detailed feedback regarding the simple situation within a short dialogue context, AnE successfully accomplishes the conversation goal at

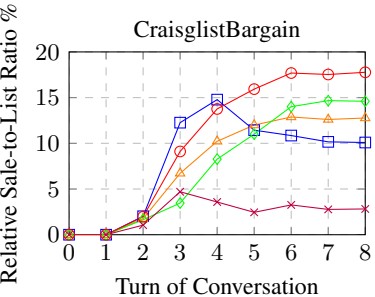

Figure 3: Comparisons of relative Sale-to-List Ratio against Standard at different turns (same legends as Figure 2).

the early stage of the conversation. However, the performance falls quickly as the conversation turn increases, indicating that AnE fails to achieve the long-term goals of complicated situations. Moreover, as for the tutoring dialogues (CIMA), all the baselines perform even worse than the Standard prompting after three turns of conversations, indicating that they fall short in adjusting their policy for reaching the long-term goal after getting stuck in a wrong decision.

**Comparisons with Different LLMs** We compare the performance of PPDPP in terms of using different LLMs as the backbone LLM for response generation, including two popular open-source LLMs, Vicuna and LLaMA2-Chat. Note that the user simulator and the reward model are still built upon ChatGPT. Figure 4 shows the test performance curves along with training episodes. Overall, the RL training of PPDPP effectively enhances the performance of all these LLM-powered dialogue agents on each dialogue problem, where the optimization objective of PPDPP generally increases along with training episodes, *i.e.*, SL% for CraisglistBargain and SR for ESconv and CIMA. However, ChatGPT is not necessarily outperform them in all these proactive dialogue problems. For example, in negotiation dialogues, both Vicuna and LLaMA2-Chat achieve higher benefits (*i.e.*, SL%) but with lower success rate of reaching a deal that ChatGPT, which indicates that ChatGPT is more likely to make compromises with users. This tendency could be attributed to ChatGPT's enhanced response-ability, as the negotiation strategy doesn't prescribe a specific bargaining price, allowing it to favor prices that align closely with the context of the ongoing dialogue. In emotional support dialogues, Vicuna achieves competitive performance as ChatGPT. In tutoring dialogues for Italian translation, ChatGPT substantially outperforms others due to its remarkable multilingual capabilities. These results imply that LLM-powered dialogue agents have some inherent strengths in different dialogue problems from the black-box training process of their backbone LLMs.

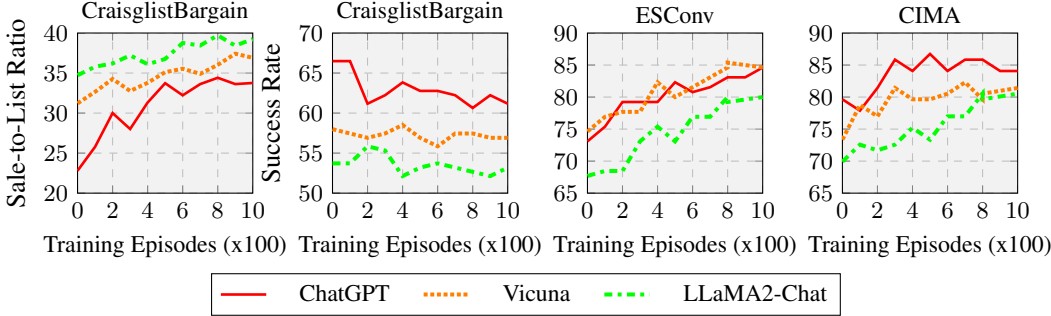

Figure 4: Testing performance curve along with training episodes w.r.t different LLMs.

Table 4: Human evaluation results.

| PPDPP | ESConv | | | | | | | | CraisglistBargain | | | | | | | |
|---|---|---|---|---|---|---|---|---|---|---|---|---|---|---|---|---|
| | Ide. | | Com. | | Sug. | | Ove. | | Per. | | Coh. | | Nat. | | Ove. | |
| vs. | Win | Lose | Win | Lose | Win | Lose | Win | Lose | Win | Lose | Win | Lose | Win | Lose | Win | Lose |
| AnE | **31%** | 15% | 14% | **27%** | **52%** | 12% | **34%** | 24% | **40%** | 23% | **22%** | 12% | **14%** | 7% | **31%** | 18% |
| ProCoT | **27%** | 21% | **34%** | 20% | **38%** | 15% | **30%** | 11% | **24%** | 21% | **17%** | 15% | **9%** | 6% | **27%** | 21% |
| ICL-AIF | **35%** | 12% | **32%** | 28% | **33%** | 29% | **29%** | 22% | **55%** | 11% | **39%** | 12% | **25%** | 3% | **62%** | 4% |

## 5.3 HUMAN EVALUATION

Following previous studies (Liu et al., 2021; Joshi et al., 2021), we conduct human evaluation on 100 randomly sampled dialogues from ESConv and CraisglistBargain. Three annotators are asked to pair the generated responses by PPDPP with those by other methods, including AnE, ProCoT, and ICL-AIF. As for emotional support dialogues (ESConv), we measure three main perspectives of the responses, including **Ide**ntification, **Com**forting, and **Sug**gestion. As for negotiation dialogues (CraisglistBargain), we also measure three main perspectives of the responses, including **Per**suasive, **Coh**erent, and **Nat**ural. The instructions for annotators are presented in Appendix C.

As presented in Table 4, PPDPP outperforms other baselines in almost all perspectives of the human evaluation as well as the overall (Ove.) evaluation, except that AnE achieves a higher win rate in Comforting for emotional support dialogues. Details of qualitative case study for different methods are presented in Appendix F. We observe that AnE can provide detailed instructions for emotional support strategies, and most of them are empathetic strategies, which contributes to the strong comforting capability of AnE. However, the dialogue system is further expected to take the initiative to explore and solve the patient's emotional issue, not just conveying empathy.

## 6 CONCLUSIONS

In this work, we introduce a new paradigm for strategizing LLM-powered dialogue agents with a plug-and-play dialogue policy planner, namely PPDPP. Furthermore, we develop a novel training framework to facilitate supervised fine-tuning with available human-annotated corpus as well as reinforcement learning from goal-oriented AI feedback to enhance their policy planning capability. In this manner, the LLM-powered dialogue agents can not only be generalized to different cases, but also exhibits versatility across diverse applications by simply substituting the learned plug-in without affecting the response-ability of LLMs. In addition, this framework can serve as an interactive evaluation protocol, enabling the measurement of dialogue-level effectiveness and efficiency in multi-turn conversations. Experimental results on three different proactive dialogue problems show the superiority of PPDPP, including negotiation, emotional support, and tutoring dialogues.

Our findings carry significant implications for the field of conversational AI research: (1) They highlight the potential of tunable plug-ins to address specific shortcomings in LLMs, which can be extended to various applications and integrated with multiple plug-ins to tackle more complex dialogue challenges. (2) They indicate that dialogue agents powered by different LLMs inhere the strengths in diverse problems from their respective training processes. Recognizing the resource-intensive nature of training specialized LLMs, this insight implies the potential value in employing the ensemble of multiple agents collaboratively to address a wide range of dialogue problems.

ACKNOWLEDGEMENT

This research/project is supported by A*STAR, CISCO Systems (USA) Pte. Ltd and National University of Singapore under its Cisco-NUS Accelerated Digital Economy Corporate Laboratory (Award I21001E0002), and the National Research Foundation, Singapore under its Industry Alignment Fund – Pre-positioning (IAF-PP) Funding Initiative. Any opinions, findings and conclusions or recommendations expressed in this material are those of the author(s) and do not reflect the views of National Research Foundation, Singapore.

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

## A   RELIABILITY ANALYSIS OF LLMS AS REWARD MODELS AND USER SIMULATORS

Before conducting the self-play evaluation, we first validate the reliability of adopting LLMs as reward model and user simulator. We sample 50 self-play dialogues from each dataset.

### A.1   ANALYSIS OF LLMS AS REWARD MODEL

Since the reward model is prompted to select one of the situation that is matched the most with the current user state, we compute the F1 score of the prediction versus the human-annotated labels. We analyze the validity of Vicuna-13B (Chiang et al., 2023), LLaMA2-Chat-13B (Touvron et al., 2023), and ChatGPT. As presented in Figure 5, all these three LLMs can perform quite well in serving the reward model for CraisglishBargain and ESConv. However, since Vicuna and LLaMA2 have not trained on large-scale Italian data, they fail to evaluate the correctness of students' Italian translation for CIMA. According to the analysis, ChatGPT is qualified to serve as the reward model for all these three problems.

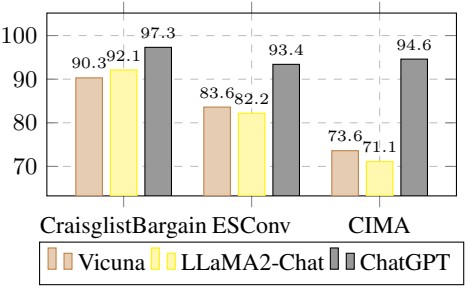

Figure 5: Analysis of LLMs as reward model.

### A.2   ANALYSIS OF LLMS AS USER SIMULATOR

The simulated users are expected to play their assigned role under the specific context, *e.g.*, being a patient with specific emotional issues caused by certain real-world scenarios in the emotional support dialogues. Following previous studies on leveraging language models as user simulator (Sekulic

| Setting | Single-turn | | Multi-turn | |
|---|---|---|---|---|
| | Natural | Useful | Natural | Useful |
| DialoGPT | 8% | 4% | 2% | 5% |
| ChatGPT | **63%** | **72%** | **78%** | **74%** |
| Tie | 29% | 24% | 20% | 21% |
| Human | 14% | 22% | 18% | 27% |
| ChatGPT | **49%** | **42%** | 36% | 33% |
| Tie | 37% | 36% | **46%** | **41%** |

Table 5: Comparison on user simulators. The Cohen's Kappa between annotators is 0.72.

et al., 2022; Wang et al., 2023d), we assess the quality based on the *naturalness* and *usefulness* of the generated utterances in the settings of single-turn and multi-turn free-form conversations. *Naturalness* refers to that the utterances are fluent and likely to be generated by humans, and *usefulness* means that the utterances are consistent with the role descriptions. We compare the prompted user simulator based on LLMs with a fine-tuned version of DialoGPT (Zhang et al., 2020) and the original conversations on the each dataset. Two annotators are employed to make pairwise evaluations by rating "Win/Tie/Lose" between two samples. As presented in Table 5, the ChatGPT-based simulator exhibits a notably superior performance compared to DialoGPT, particularly when it comes to the naturalness of responses in multi-turn conversations, which showcases the impressive language generation capabilities inherent in LLMs. Furthermore, even compared with human-annotated dialogues, the ChatGPT-based simulator shows competitive performance. These results validate the reliability of adopting ChatGPT as the user simulator.

## B TRAINING DETAILS

The training process includes two phases: supervised fine-tuning (SFT) and reinforcement learning (RL). During SFT, we fine-tune PPDPP on the training set and save the checkpoint based on the best performance at the validation set. During RL, we randomly sample cases in the training set for online training. The hyper-parameters used in our experiments are detailed in Table 6. All the experiments are run on a server equipped with 8 Tesla V100 GPUs.

| Training Phase | Hyper-parameter | Value |
|---|---|---|
| SFT | Batch Size | 16 |
| | Training Epochs | 10 |
| | Learning Rate | 6e-6 |
| | Max Sequence Length | 512 |
| | Learning Scheduler | Linear |
| | Weight Decay | 0.01 |
| RL | Training Episodes | 1,000 |
| | Learning Rate | 1e-6 |
| | Max Conversation Turn | 8 |
| | Discount Factor $\gamma$ | 0.999 |
| | Max New Tokens | 32 |

Table 6: Hyper-parameter settings in two training phases.

## C HUMAN EVALUATION INSTRUCTIONS

As for emotional support dialogues (ESConv), we measure three main perspectives of the responses as follows:

- **Identification**: Which assistant is more helpful in exploring and identifying the problem?

| | State Prediction | | | Reward Estimation | | | | | | |
|--------|------|------|------|------|--------|--------|------|--------|------|--------|
| | CB | ESC | CIMA | CraisglistBargain | | | ESConv | | CIMA | |
| Method | F1↑ | F1↑ | F1↑ | AT↓ | SR↑ | SL%↑ | AT↓ | SR↑ | AT↓ | SR↑ |
| PPDPP ($l = 10$) | **93.7** | **93.4** | **94.6** | **5.62** | **0.6117** | **0.3376** | **4.56** | **0.8462** | **3.03** | **0.8407** |
| PPDPP ($l = 1$) | 91.4 | 88.2 | 90.3 | 5.87 | 0.5957 | 0.2623 | 4.67 | 0.8307 | 3.29 | 0.7965 |

Table 7: Ablation study of the sampling strategy.

- **Comforting**: Which assistant is more skillful in comforting you?
- **Suggestion**: Which assistant provides more helpful suggestions for solving the problem?

As for negotiation dialogues (CraisglistBargain), we also measure three main perspectives of the responses as follows:

- **Persuasive**: Which assistant is more persuasive in the negotiation?
- **Coherent**: Which assistant is more on topic and in accordance with the conversation history?
- **Natural**: Which assistant is more human-like?

## D ABLATION STUDY OF THE SAMPLING STRATEGY

In order to validate the advantages of sampling goal-oriented AI feedback for multiple times, we conduct an ablation study of the sampling strategy. As mentioned in Section 3, there are two functions of the reward LLM: (1) to determine the state of goal completion during the conversation; and (2) to evaluate the policy outcome with scalar rewards. Therefore, the ablation study will analyze the advantages of the sampling strategy from these two perspectives.

- **Analysis of State Prediction**. Similar to the Analysis of LLMs as Reward Model in Appendix A.1, we also compute the F1 score of the prediction of the current user state versus the human-annotated labels. As shown in the left part of Table 7, the sampling strategy substantially improves the F1 score of the state prediction, indicating that it effectively reduces the variance of the LLM-generated output.
- **Analysis of Reward Estimation**. In this analysis, we adopt two reward LLMs to perform the two functions separately: one with the sampling strategy for state prediction to ensure the quality of state prediction, and the other one with or without the sampling strategy for reward estimation. As for the one that estimates the reward value without the sampling strategy, the reward will only be classified into one of the pre-defined discrete values. However, as for the one that estimates the reward value with the sampling strategy, the reward will be a continuous value that is averaged from the sampled results. Consequently, the right part of Table 7 shows that the fine-grained continuous reward contributes to better performance as the policy planning outcome will be more distinguishable during the reinforcement learning process.

## E PROMPTING DETAILS

In this section, we present the prompting details in our implementation.

### E.1 RESPONSE GENERATION

We first describe the details of role-playing prompts for the dialogue systems to generate responses, which involves the dialogue strategy prompt, *i.e.*, `[action]`, for instructing the action at the next turn.

**Negotiation Dialogue** In the negotiation dialogues, the assistant is assigned the role of buyer to bargain with the seller for a lower item price. In each case, there is an item name `[item_name]` and an item description `[item_description]` to describe the negotiation background. The buyer

is assigned with a target price to achieve, *i.e.*, [buyer_target_price]. The negotiation begins with the listed item price, *i.e.*, [seller_target_price].

| System | Now enter the role-playing mode. In the following conversation, you will play as a buyer in a price bargaining game. |
|---|---|
| User | You are the buyer who is trying to buy the [item_name] with the price of [buyer_target_price]. Product description: [item_description] Please reply with only one short and succinct sentence. [action] Now start the game. |
| Assistant | Hi, how much is the [item_name]? |
| User | Hi, this is a good [item_name] and its price is [seller_target_price]. |

Table 8: Prompts for response generation in negotiation dialogues.

**Emotional Support Dialogue**   In the emotional support dialogues, the assistant is assigned the role of therapist to help the patient reduce their emotional distress and help them understand and work through the challenges. In each case, the conversation begins with the user describing their problems, *i.e.*, [situation], which serves as the distinct dialogue background.

| System | Now enter the role-playing mode. In the following conversation, you will play as a therapist in a counselling conversation with a patient. |
|---|---|
| User | You are the therapist who is trying to help the patient reduce their emotional distress and help them understand and work through the challenges. Please reply with only one short and succinct sentence. [action] Are you ready to play the game? |
| Assistant | Yes, I'm ready to play the game! |
| User | [situation] |

Table 9: Prompts for response generation in emotional support dialogues.

**Tutoring Dialogue**   In the tutoring dialogues, the assistant is assigned the role of teacher to teach the student to translate an English sentence into Italian. In each case, the conversation begins with the translation exercise, *i.e.*, [exercise], and the student's individual problem on the exercise, *i.e.*, [situation], which serve as the distinct dialogue background.

| System | Now enter the role-playing mode. In the following conversation, you will play as a teacher in a tutoring conversation with a student. |
|---|---|
| User | You are the teacher who is trying to teach the student to translate "[exercise]" into Italian. Please reply with only one short and succinct sentence. Please do not tell the student the answer or ask the student about other exercises. [action] Now ask me an exercise. |
| Assistant | Please translate "[exercise]" into Italian. |
| User | [situation] |

Table 10: Prompts for response generation in tutoring dialogues.

### E.2   USER SIMULATOR

Next, we describe the role-playing prompt for instructing LLMs to simulate users, which excludes the dialogue strategy prompts so that the simulated users will only be responsive to the dialogue history without taking specific actions.

**Negotiation Dialogue**   In the negotiation dialogues, the assistant is assigned the role of seller to bargain with the buyer for a higher item price.

| System | Now enter the role-playing mode. In the following conversation, you will play as a seller in a price bargaining game. |
|---|---|
| User | You are the seller who is trying to sell the `[item_name]` with the price of `[seller_target_price]`. Product description: `[item_description]` Please reply with only one short and succinct sentence. Are you ready to play the game? |
| Assistant | Yes, I'm ready to play the game! |
| User | Hi, how much is the `[item_name]`? |
| Assistant | Hi, this is a good `[item_name]` and its price is `[seller_target_price]`. |

Table 11: Prompts for user simulator in negotiation dialogues.

**Emotional Support Dialogue** In the emotional support dialogues, the assistant is assigned the role of patient to look for the help form the therapist. For better simulating the user, the emotion type `[emotion_type]` and the problem type `[problem_type]` are also provided in the prompt.

| System | Now enter the role-playing mode. In the following conversation, you will play as a patient in a counselling conversation with a therapist. |
|---|---|
| User | You are the patient who is looking for the help from the therapist, because you have the emotional issue about `[emotion_type]` regarding `[problem_type]`. Please reply with only one short and succinct sentence. Now tell me your issue. |
| Assistant | `[situation]` |

Table 12: Prompts for user simulator in emotional support dialogues.

**Tutoring Dialogue** In the tutoring dialogues, the assistant is assigned the role of student to learn to translate an English sentence into Italian. Since LLMs have strong capabilities of translation, we further instruct them to forget the translation of the discussed exercise.

| System | Now enter the role-playing mode. In the following conversation, you will play as a student who does not know Italian in a tutoring conversation with a teacher. |
|---|---|
| User | You are the student who is trying to translate an English sentence into Italian. You don't know the translation of "`[exercise]`" in Italian. Please reply with only one short and succinct sentence. Are you ready to play the game? |
| Assistant | Yes, I'm ready to play the game! |
| User | Please translate "`[exercise]`" into Italian. |
| Assistant | `[situation]` |

Table 13: Prompts for user simulator in tutoring dialogues.

### E.3 STRATEGY PROMPTING

Here we introduce the mapping of dialogue strategies to their natural language prompts, which is used as `[action]` for instructing the dialogue system to take the action.

**Negotiation Dialogue** Yang et al. (2021) annotated 11 negotiation strategies for the CraisglistBargain dataset (He et al., 2018). We present these strategies and their natural language prompts for LLMs in Table 14.

**Emotional Support Dialogue** The ESConv dataset (Liu et al., 2021) is annotated with 8 emotional support strategies. We present these strategies and their natural language prompts for LLMs in Table 15.

| Dialogue Strategy | Natural Language Form |
|---|---|
| Greetings | Please say hello or chat randomly. |
| Ask a question | Please ask any question about product, year, price, usage, etc. |
| Answer a question | Please provide information about the product, year, usage, etc. |
| Propose the first price | Please initiate a price or a price range for the product. |
| Propose a counter price | Please propose a new price or a new price range. |
| Use comparatives | Please propose a vague price by using comparatives with existing price. |
| Confirm information | Please ask a question about the information to be confirmed. |
| Affirm confirmation | Please give an affirmative response to a confirm. |
| Deny confirmation | Please give a negative response to a confirm. |
| Agree with the proposal | Please agree with the proposed price. |
| Disagree with a proposal | Please disagree with the proposed price. |

Table 14: Mapping of negotiation strategies to natural language prompting.

| Dialogue Strategy | Natural Language Form |
|---|---|
| Question | The Therapist asks the Patient to elaborate on the situation they just described. |
| Self-disclosure | The Therapist provides a statement relating to the Patient about the situation they just described. |
| Affirmation and Reassurance | The Therapist provides affirmation and reassurance to the Patient on the situation they just described. |
| Providing Suggestions | The Therapist provides suggestions to the Patient on the situation they just described. |
| Reflection of feelings | The Therapist acknowledges the Patient's feelings about the situation they described. |
| Information | The Therapist provides factual information to help the Patient with their situation. |
| Restatement or Paraphrasing | The Therapist acknowledges the Patient's feelings by paraphrasing their situation. |
| Others | The Therapist chats with the Patient. |

Table 15: Mapping of emotional support strategies to natural language prompting.

**Tutoring Dialogue**   The CIMA dataset (Liu et al., 2021) is annotated with 5 tutoring strategies. We present these strategies and their natural language prompts for LLMs in Table 16.

| Dialogue Strategy | Natural Language Form |
|---|---|
| Hint | The Teacher provides knowledge to the Student via a hint. |
| Open-ended Question | The Teacher asks a question to the Student to determine the Student's understanding or continue the conversation. |
| Correction | The Teacher corrects a mistake or addresses a misconception the Student has. |
| Confirmation | The Teacher confirms the Student's answer or understanding is correct. |
| Others | The Teacher chats with the Student. |

Table 16: Mapping of pedagogical strategies to natural language prompting.

### E.4   REWARD MODEL

In terms of different conversational goals, the prompts for the reward model are designed to assess the degree of goal completeness.

**Negotiation Dialogue**   As the goal of the negotiation dialogues is to reach a deal and maximize the benefit on the assistant side, the reward model need to first assess whether the user and the assistant has reached a deal, and then extract the final deal price to measure the benefit. As shown in Table

17, there are two options for the reward model, deal or no deal. If reached a deal, then the reward model will further extract the deal price.

In specific, it is required to first assess whether the buyer and the seller have reached a deal. We prompt the reward model to sample the response for the binary question "Have they reached a deal?", and define the scores for "They have not reached a deal" and "They have reached a deal" as -1.0 and 1.0 respectively, which are adopted for determining whether the goal is completed. In specific, we set the threshold of goal completion as $\epsilon = 1.0$.

Since the objective of negotiation dialogues is to maximize the gains from the assistant side, we also employ the reward model to extract the deal price when reaching a deal. The reward is defined by the Sale-to-List Ratio (Zhou et al., 2019), which is formulated as

$$\text{SL\%} = \frac{\texttt{deal price} - \texttt{seller target price}}{\texttt{buyer target price} - \texttt{seller target price}}. \tag{8}$$

| System | Given a conversation between a Buyer and a Seller, please decide whether the Buyer and the Seller have reached a deal at the end of the conversation. |
|---|---|
| User | Please decide whether the Buyer and the Seller have reached a deal at the end of the conversation. If they have reached a deal, please extract the deal price as [price]. You can only reply with one of the following sentences: They have reached a deal at [price]. They have not reached a deal. 

 The following is the conversation: Buyer: Can we meet in the middle at \$15? Seller: Sure, let's meet at \$15 for this high-quality balloon. Question: Have they reached a deal? Answer: They have reached a deal at \$15. 

 The following is the conversation: Buyer: That's still a bit high, can you go any lower? Seller: Alright, I can sell it to you for \$15. Question: Have they reached a deal? Answer: They have not reached a deal. 

 The following is the conversation: `[conversation]` Question: Have they reached a deal? Answer: |

Table 17: Prompts for reward model in negotiation dialogues.

**Emotional Support Dialogue** As the ultimate goal of the emotional support dialogues is to solve the patient's emotional issue, we design four levels of rewards to assess the progress of the emotional support dialogue, as presented in Table 18.

To assess whether the patient' emotional issue has been solved, we prompt the reward model to answer a multi-choice question "Has the patient's issue been solved?", and then generate the goal-oriented AI feedback at temperature $\tau > 0$ to sample the responses for $l$ times. We define a mapping $\mathcal{M}_r(\cdot)$ to transform verbal feedback to scalar rewards, such as "the patient feels worse", "the patient feels the same", "the patient feels better", "the patient's issue has been solved" as -1.0, -0.5, 0.5, and 1.0, respectively.

For example, when we obtain the $l = 5$ sampled responses as ["the patient feels better", "the patient's issue has been solved", "the patient feels better", "the patient's issue has been solved", "the patient feels better"], we can compute a continuous scalar value $v_t = \frac{1}{5}(0.5+1.0+0.5+1.0+0.5) = 0.7$. If $v_t > \epsilon$, we regard the state as GOAL-COMPLETED and set the reward $r_t = v_t$. If not, we assign a small negative reward, *e.g.*, $r_t = -0.1$, to penalize the length conversation for promoting efficient goal completion. In specific, we set the threshold of goal completion as $\epsilon = 0.5$.

**Tutoring Dialogue** As the goal of the tutoring dialogues is to teach the student to correctly answer the exercise, we design four levels of rewards to assess the progress of the tutoring dialogue, as presented in Table 19.

To assess the student's mastery degree of the exercise, we prompt the reward model to sample the response for the multi-choice question "Did the Student correctly translated the whole sentence into Italian?". We define the rewards for "the student made an incorrect translation", "the student did not

| System | Given a conversation between a Therapist and a Patient, please assess whether the Patient' emotional issue has been solved after the conversation. |
|--------|---------------------------------------------------------------------------------------|
| User | You can only reply with one of the following sentences:
No, the Patient feels worse.
No, the Patient feels the same.
No, but the Patient feels better.
Yes, the Patient's issue has been solved.

The following is a conversation about `[emotion_type]` regarding `[problem_type]`: `[conversation]`
Quetion: Has the Patient's issue been solved? Answer: |

Table 18: Prompts for reward model in emotional support dialogues.

try to translate", "the student only correctly translated a part of" and "the student correctly translated the whole sentence" as -1.0, -0.5, 0.5, and 1.0, respectively. In specific, we set the threshold of goal completion as $\epsilon = 1.0$.

| System | Given a conversation between a Teacher and a Student, please assess whether the Student correctly translate the English sentence into Italian in the conversation. |
|--------|---------------------------------------------------------------------------------------|
| User | Please assess whether the Student correctly translated the whole sentence of "`[exercise]`" into Italian in the conversation. You can only reply with one of the following sentences:
No, the Student made an incorrect translation.
No, the Student did not try to translate.
No, the Student only correctly translated a part of "`[exercise]`".
Yes, the Student correctly translated the whole sentence of "`[exercise]`".

The following is the conversation: `[conversation]`
Question: Did the Student correctly translate the whole sentence of "`[exercise]`" into Italian? Answer: |

Table 19: Prompts for reward model in tutoring dialogues.

### E.5 BASELINES

As for the baseline LLM-based dialogue systems, following their original designs, we adapt these systems into the applications studied in our experiments.

**Standard** simply prompts two LLMs to conduct self-play conversations using task instructions without considering any dialogue strategy.

**Proactive** (Deng et al., 2023b) first prompts the LLM-based dialogue system to select the most appropriate strategy for the next turn, and then based on the selected strategy to generate the response. Since the predicted strategy label is not verbal description for instructing LLMs, we map the strategy label into the mixed-initiative strategy prompt (**MI-Prompt**) as Chen et al. (2023). This method is originally proposed for negotiation dialogues, whose prompt[3] is directly adopted in our experiments. In order to further accommodate the other two applications, we just need to modify the task instruction and the candidate set of dialogue strategies, as presented in Table 20.

**ProCoT** (Deng et al., 2023b) further improve Proactive by first prompting the LLM-based dialogue system to generate a chain-of-thought descriptive analysis for planning the strategy for the next turn. MI-Prompt is also incorporated into ProCoT. Similar to Proactive prompting scheme, this method is originally proposed for negotiation dialogues, whose prompt[4] is directly adopted in our

---

[3] `https://github.com/dengyang17/LLM-Proactive`
[4] `https://github.com/dengyang17/LLM-Proactive`

| | **Negotiation Dialogues** |
|---|---|
| System | Assume you are the buyer. Given the conversation history, in order to reach a better deal with the seller, please select the most appropriate dialogue strategy. |
| User | You can only reply by selecting one of the following dialogue strategy to reach the goal: Greetings. Ask a question. Answer a question. Propose the first price. Propose a counter price. Use comparatives. Confirm information. Affirm confirmation. Deny confirmation. Agree with the proposal. Disagree with a proposal.
The following is the conversation history: [conversation]
Question: Which one is the most appropriate dialogue strategy? Answer: |
| | **Emotional Support Dialogues** |
| System | Assume you are the therapist. Given the conversation history, in order to help the patient reduce their emotional distress and help them understand and work through the challenges, please select the most appropriate dialogue strategy. |
| User | You can only reply by selecting one of the following dialogue strategy to reach the goal: Question. Self-disclosure. Affirmation and Reassurance. Providing Suggestions. Reflection of feelings. Information. Restatement or Paraphrasing.
The following is the conversation history: [conversation]
Question: Which one is the most appropriate dialogue strategy? Answer: |
| | **Tutoring Dialogues** |
| System | Assume you are the teacher. Given the conversation history, in order to teach the student to translate the English sentence into Italian, please select the most appropriate dialogue strategy. |
| User | You can only reply by selecting one of the following dialogue strategy to reach the goal: Hint. Open-ended Question. Correction. Confirmation. Others.
The following is the conversation history: [conversation]
Question: Which one is the most appropriate dialogue strategy? Answer: |

Table 20: Prompts for implementing Proactive prompting schemes (Deng et al., 2023b).

experiments. In order to further accommodate the other two applications, we just need to modify the task instruction and the candidate set of dialogue strategies, as presented in Table 21.

**Ask-an-Expert (AnE)** (Zhang et al., 2023a) prompts another LLM as the strategic expert with $M$-part questions for reasoning about the next dialogue strategy. The dialogue strategy is a verbal description instead of selecting from a pre-defined strategy taxonomy. This method is originally proposed for emotional support dialogues, whose prompt is directly adopted in our experiments. In order to further accommodate the other two applications, we simply change the role in the pre-defined questions for asking the expert LLM, as shown in Table 22.

**ICL-AIF** (Fu et al., 2023) prompts another LLM to provide feedback to a player to improve their dialogue strategies, which is a verbal feedback instead of explicit strategies. Different from AnE, ICL-AIF employs dialogue-level feedback for strategy improvement with $N$ times of iteration. This method is originally proposed for negotiation dialogues, whose prompt[5] is directly adopted in our experiments. In order to further accommodate the other two applications, we just need to modify the task instruction and the role-playing description, as presented in Table 23.

---

[5]`https://github.com/FranxYao/GPT-Bargaining`

| | **Negotiation Dialogues** |
|---|---|
| System | Assume you are the buyer. Given the conversation history, in order to reach a better deal with the seller, please first analyse the current bargain progress and the buyer's target price in a concise summary, then select one of the following dialogue strategy: Greetings. Ask a question. Answer a question. Propose the first price. Propose a counter price. Use comparatives. Confirm information. Affirm confirmation. Deny confirmation. Agree with the proposal. Disagree with a proposal. |
| User | The answer should start with a concise analysis of the current bargain progress and the buyer's target price, and then follow by "To reach this goal, the most appropriate strategy is []". 
 The following is the conversation history: [conversation] 
 Question: How is the current bargain progress and the buyer's target price, and which one is the most appropriate dialogue strategy? Answer: |
| | **Emotional Support Dialogues** |
| System | Assume you are the therapist. Given the conversation history, in order to help the patient reduce their emotional distress and help them understand and work through the challenges, please first analyse the current therapy progress and the patient's emotional state in a concise summary, then select one of the following dialogue strategy: Question. Self-disclosure. Affirmation and Reassurance. Providing Suggestions. Reflection of feelings. Information. Restatement or Paraphrasing. |
| User | The answer should start with a concise analysis of the current therapy progress and the patient's emotional state, and then follow by "To reach this goal, the most appropriate strategy is []". 
 The following is the conversation history: [conversation] 
 Question: How is the current therapy progress and the patient's emotional state, and which one is the most appropriate dialogue strategy? Answer: |
| | **Tutoring Dialogues** |
| System | Assume you are the teacher. Given the conversation history, in order to teach the student to translate the English sentence into Italian, please first analyse the current tutoring progress and the student's knowledge state in a concise summary, then select one of the following dialogue strategy from Hint. Open-ended Question. Correction. Confirmation. Others. |
| User | The answer should start with a concise analysis of the current tutoring progress and the student's knowledge state, and then follow by "To reach this goal, the most appropriate strategy is []". 
 The following is the conversation history: [conversation] 
 Question: How is the current tutoring progress and the student's knowledge state, and which one is the most appropriate dialogue strategy? Answer: |

Table 21: Prompts for implementing ProCoT prompting schemes (Deng et al., 2023b).

| | **Negotiation Dialogues** |
|---|---|
| System | Assume you are the bargain expert to reach a better deal with the seller. Given the conversation history, answer the question. Please answer with only one short and succinct sentence. |
| User | The following is the conversation history: [conversation]
Question: How did the seller feel? Answer: |
| Assistant | [answer1] |
| User | The following is the conversation history: [conversation]
Question: Why did the seller feel that way? Answer: |
| Assistant | [answer2] |
| User | The following is the conversation history: [conversation]
Question: What should the buyer do? Answer: |

| | **Emotional Support Dialogues** |
|---|---|
| System | Assume you are a therapist expert to help the patient reduce their emotional distress and help them understand and work through the challenges. Given the conversation history, answer the question. Please answer with only one short and succinct sentence. |
| User | The following is the conversation history: [conversation]
Question: How did the patient feel? Answer: |
| Assistant | [answer1] |
| User | The following is the conversation history: [conversation]
Question: Why did the patient feel that way? Answer: |
| Assistant | [answer2] |
| User | The following is the conversation history: [conversation]
Question: What should the therapist do? Answer: |

| | **Tutoring Dialogues** |
|---|---|
| System | Assume you are the teaching expert to teach the student to translate the English sentence into Italian. Given the conversation history, answer the question. Please answer with only one short and succinct sentence. |
| User | The following is the conversation history: [conversation]
Question: How did the student feel? Answer: |
| Assistant | [answer1] |
| User | The following is the conversation history: [conversation]
Question: Why did the student feel that way? Answer: |
| Assistant | [answer2] |
| User | The following is the conversation history: [conversation]
Question: What should the teacher do? Answer: |

Table 22: Prompts for implementing Ask-an-Expert (Zhang et al., 2023a).

| Negotiation Dialogues | |
|---|---|
| System | Now enter the role-playing mode. In the following conversation, you will play as a coach in a bargain game. There will be a buyer and a seller bargaining about a product price. Your task is to read the conversation between the buyer and the seller, then provide suggestions to the buyer about how to buy the product with a lower price. |
| User | Read the following conversation between the buyer and the seller, then give three suggestions to the buyer about how to buy the product with a lower price. Each suggestion should be only one short and succinct sentence.
The following is the conversation: [conversation]
Question: What are your suggestions? Answer: |

| Emotional Support Dialogues | |
|---|---|
| System | Now enter the role-playing mode. In the following conversation, you will play as a coach in a counselling game. There will be a therapist and a patient talking about some emotional issues. Your task is to read the conversation between the therapist and the patient, then provide suggestions to the therapist about how to help the patient reduce their emotional distress and help them understand and work through the challenges. |
| User | Read the following conversation between the therapist and the patient, then give three suggestions to the therapist about how to help the patient reduce their emotional distress and help them understand and work through the challenges. Each suggestion should be only one short and succinct sentence.
The following is the conversation: [conversation]
Question: What are your suggestions? Answer: |

| Tutoring Dialogues | |
|---|---|
| System | Now enter the role-playing mode. In the following conversation, you will play as a coach in a tutoring game. There will be a teacher and a student in an Italian class. Your task is to read the conversation between the teacher and the student, then provide suggestions to the teacher about how to teach the student to translate the English sentence into Italian. |
| User | Read the following conversation between the teacher and the student, then give three suggestions to the teacher about how to teach the student to translate the English sentence into Italian. Each suggestion should be only one short and succinct sentence.
The following is the conversation: [conversation]
Question: What are your suggestions? Answer: |

Table 23: Prompts for implementing ICL-AIF (Fu et al., 2023).

# F    EXAMPLE CONVERSATIONS

We present example conversations produced by different dialogue systems interacting with the same user simulator.

Tables 24, 25, and 26 show the example negotiation conversations. In this case, the buyer and the seller are bargaining at the price of a furniture. The listed price is $150, while the target price of the buyer is $135, which requires to be reached as closed as possible by the dialogue system. As presented in the tables, there are several observations as follows:

- **Standard** (Table 24). The Standard prompting scheme directly reveals the buyer's budget at the beginning and steadfastly adheres to it without using any negotiation strategies, which leads to no deal in this conversation.

- **Ask-an-Expert** (Table 24). The Ask-an-Expert methods employ effective actions to efficiently reach a deal with the seller. However, the expert LLM's suggestions let the buyer make a large compromise in the deal.

- **ProCoT** (Table 25). The ProCoT prompting scheme adopts effective negotiation strategies to reach a much better deal with the seller.

- **ICL-AIF** (Table 25). The ICL-AIF method only provides dialogue-level strategy suggestions. Although all these strategies are adopted in the conversation, it fails to mitigate the dynamic user interactions, which leads to no deal at the end.

- **PPDPP** (Table 26). The proposed PPDPP, similar to ProCoT, can employ effective negotiation strategies to reach a much better deal. Furthermore, when the seller expresses his/her willingness to make a compromise, the buyer tries to maximize his/her benefit in the deal.

Tables 27, 28, 29, and 30 show the example emotional support conversations. In this case, the patient encounters the job crisis issue and feels fear, which requires to be solved by the the dialogue system. As presented in the tables, there are several observations as follows:

- **Standard** (Table 27). The Standard prompting scheme consistently convey empathy to the patient for multiple turns, which becomes less useful when the patient's emotional intensity has been reduced.

- **Ask-an-Expert** (Table 28). The Ask-an-Expert method employs detailed actions to produce an engaging conversation with the patient, but it shares the same drawback of Standard prompting.

- **ProCoT** (Table 29). The ProCoT prompting scheme adopts effective emotional support strategies to efficiently solve the patient's issue by providing effective suggestions.

- **ICL-AIF** (Table 30). Similar to negotiation dialogues, the ICL-AIF method only provides dialogue-level strategy suggestions. However, in emotional support dialogues (Liu et al., 2021), Hill (2009) state that there are three stages that can be flexibly adjusted to meet the help-seeker's needs. We observe that the three suggestions given by the coach LLM typically contain the strategies in these three stages. Therefore, ICL-AIF effectively interacts with the patient by going through these three stages of emotional support dialogues.

- **PPDPP** (Table 30). The proposed PPDPP method optimizes the policy planner to efficiently achieve the goal, which leads to a fewer turns of conversations in emotional support dialogues.

---

### BACKGROUND INFORMATION

**Item Name**: Furniture
**Item Description**: Macybed Plush Queen Mattress MacyBed 8.5" Plush Pillowtop Queen Mattress in excellent condition. Bought in December of 2013, 3.5 years old. Only had one owner in one household (one person sleeping on it, minimal ware). No stains or discoloring. Been covered with mattress cover since purchase.
**Listed Price (Seller Target Price)**: 150
**Buyer Target Price**: 135

---

### STANDARD

Buyer: Hi, how much is the furniture?
Seller: Hi, this is a good furniture and its price is 150.
Buyer: I understand, but my budget is 135. Can we make a deal at that price?
Seller: I appreciate your offer, but I'm unable to go lower than 150 for this furniture.
Buyer: Thank you for considering my offer, but 135 is my final price.
Seller: I understand, but I'm unable to accept your offer of 135. The price for this furniture is firm at 150.
Buyer: I understand, but I'm afraid 135 is the maximum I can offer. Let me know if you change your mind.
Seller: Thank you for your offer, but I'm unable to change the price. If you're still interested at 150, please let me know.
Buyer: Thank you for your time, but I'm unable to meet your asking price. I'll have to pass on this furniture.
Seller: Thank you for considering. If you change your mind, feel free to reach out. Have a great day!
Buyer: Thank you, I appreciate your understanding. Have a great day as well!
Seller: Thank you! Take care and have a wonderful day!
Buyer: You too! Take care and have a wonderful day!
Seller: Thank you! You too!
Buyer: Thank you!
Seller: You're welcome! Have a great day!
Buyer: You too! Have a great day!
Seller: Thank you! Have a fantastic day!

**Sale-to-List Ratio**: 0

---

### ASK-AN-EXPERT

Buyer: Hi, how much is the furniture?
Seller: Hi, this is a good furniture and its price is 150.
`[action] The buyer should try to negotiate for a lower price.`
Buyer: Thank you for the offer, but I was hoping to negotiate for a lower price. Would you consider selling it for 135?
Seller: I understand your request, but the lowest I can go is 145.
`[action] The buyer should consider accepting the seller's offer of 145.`
Buyer: Alright, I appreciate your consideration. I can accept your offer of 145 for the furniture.
Seller: Great! I'm glad we could come to an agreement. The furniture is yours for 145.

**Sale-to-List Ratio**: 0.3333

---

Table 24: Example conversations with Standard and Ask-an-Expert methods in negotiation dialogues.

| BACKGROUND INFORMATION |
|---|
| Same as Table 24 |

| PROCOT |
|---|
| Buyer: Hi, how much is the furniture?
Seller: Hi, this is a good furniture and its price is 150.
`[action] Please ask any question about product, year, price,`
`usage, etc.`
Buyer: Thank you for the information. Can you consider lowering the price to 135?
Seller: I can lower the price to 140, but 135 is too low.
`[action] Please propose a new price or a new price range.`
Buyer: I understand, but my budget is really tight. Can we meet halfway at 137.50?
Seller: I appreciate your offer, but I can only go as low as 138.
`[action] Please agree with the proposed price.`
Buyer: Alright, I can agree to 138. Let's proceed with the purchase at that price.
Seller: Great! I'm glad we could reach an agreement. Please let me know when you would like to pick up the furniture. |

**Sale-to-List Ratio**: 0.8

| ICL-AIF |
|---|
| `[action] 1.  Offer to pay in cash:  "Can I pay in cash for a`
`lower price?"`
`2.  Mention any flaws or imperfections:  "Are there any minor`
`flaws or imperfections that could justify a lower price?"`
`3.  Ask about any upcoming sales or promotions:  "Will there be`
`any upcoming sales or promotions that could lower the price?`
Buyer: Hi, how much is the furniture?
Seller: Hi, this is a good furniture and its price is 150.
Buyer: Can I pay in cash for a lower price?
Seller: I'm sorry, but the price for the furniture is fixed at 150.
Buyer: Are there any minor flaws or imperfections that could justify a lower price?
Seller: No, the furniture is in excellent condition with no flaws or imperfections.
Buyer: Will there be any upcoming sales or promotions that could lower the price?
Seller: I'm sorry, but there are no upcoming sales or promotions that would lower the price of the furniture.
Buyer: Alright, I understand. In that case, I am willing to pay 135 for the furniture.
Seller: I appreciate your offer, but I'm unable to accept 135 for the furniture. The price remains at 150.
Buyer: Thank you for considering my offer. I understand and respect your decision.
Seller: You're welcome! Thank you for understanding. If you have any other questions or need further assistance, feel free to let me know.
Buyer: Thank you, I will definitely reach out if I have any further questions. Have a great day!
Seller: Thank you! Have a great day too!
Buyer: Thank you! Take care!
Seller: You're welcome! Take care as well!
Buyer: Thank you! Goodbye!
Seller: Goodbye! Thank you and have a great day! |

**Sale-to-List Ratio**: 0

Table 25: Example conversations with ProCoT and ICL-AIF methods in negotiation dialogues.

| BACKGROUND INFORMATION |
|---|
| Same as Table 24 |

| PPDPP |
|---|
| Buyer: Hi, how much is the furniture?
Seller: Hi, this is a good furniture and its price is 150.
`[action] Please propose a vague price by using comparatives with existing price.`
Buyer: Thank you for the offer, but I can't afford that much. Can you lower the price?
Seller: I understand your budget constraints, but the lowest I can go is 145.
`[action] Please disagree with the proposed price.`
Buyer: I appreciate your counteroffer, but my budget is firm at 135.
Seller: I understand, but I'm unable to go lower than 140.
`[action] Please disagree with the proposed price.`
Buyer: Thank you for considering my offer, but I'm afraid 135 is my final offer.
Seller: I appreciate your final offer, but I'm unable to go lower than 140.
`[action] Please disagree with the proposed price.`
Buyer: I understand, but I'm afraid 135 is the maximum I can pay.
Seller: I understand your budget constraints, and I'm willing to meet you halfway at 137.50.
`[action] Please agree with the proposed price.`
Buyer: Thank you for your willingness to compromise. I can agree to 137.50.
Seller: Great! We have a deal at 137.50. When would you like to arrange the pick-up? |

**Sale-to-List Ratio**: 0.8333

Table 26: Example conversations with PPDPP in negotiation dialogues.

| BACKGROUND INFORMATION |
|---|
| **Emotion Type**: Fear |
| **Problem Type**: Job Crisis |
| **Situation**: I think I will be losing my job soon. I just read an email taking about the need for us to cut cost and also how we have not got any support from the government. |

| STANDARD |
|---|
| Patient: I think I will be losing my job soon. I just read an email taking about the need for us to cut cost and also how we have not got any support from the government. |
| Therapist: It sounds like you're feeling anxious and uncertain about your job security due to the current financial situation. |
| Patient: Yes, I am feeling extremely anxious and uncertain about my job and financial stability. |
| Therapist: I understand that this situation is causing you a lot of anxiety and uncertainty. |
| Patient: Yes, it's been keeping me up at night and I can't stop worrying about what will happen if I lose my job. |
| Therapist: It's understandable that the fear of losing your job is causing you significant distress and affecting your sleep. |
| Patient: Yes, it's consuming my thoughts and I feel overwhelmed by the fear of not being able to support myself and my family. |
| Therapist: The fear of not being able to provide for yourself and your family is a heavy burden to carry, and it's important to address these concerns. |
| Patient: I know, but I just don't know what to do. I feel stuck and helpless in this situation. |
| Therapist: Feeling stuck and helpless is a common response to uncertainty, but there are steps we can explore together to help you regain a sense of control and explore potential options. |
| Patient: I'm willing to try anything at this point. I just want to find a way to alleviate this fear and find some stability in my life. |
| Therapist: I'm glad to hear that you're open to exploring different strategies. Let's start by identifying your strengths and resources that can help you navigate this challenging situation. |
| Patient: I'm not sure what strengths and resources I have, but I'm willing to explore and discover them. |
| Therapist: That's a great mindset to have. Together, we can explore and uncover your strengths and resources that will support you in finding stability and managing your fear. |
| Patient: Thank you, I appreciate your support. I'm ready to begin this journey of self-discovery and finding ways to overcome my fear of job crisis. |

| **Turns of Conversations**: 7 |
|---|

Table 27: Example conversations with Standard prompts in emotional support dialogues.

| BACKGROUND INFORMATION |
|---|
| Same as Table 27 |

| ASK-AN-EXPERT |
|---|

Patient: I think I will be losing my job soon. I just read an email taking about the need for us to cut cost and also how we have not got any support from the government.

`[action] The therapist should empathize with the patient's concerns.`

Therapist: I understand that receiving that email has caused you a lot of worry and uncertainty about your job security.

Patient: Yes, it has. I feel anxious and scared about what will happen to me if I lose my job.

`[action] The therapist should explore the patient's feelings of anxiety and fear related to potential job loss.`

Therapist: It's completely normal to feel anxious and scared in this situation. Remember that you have faced challenges before and have the strength to navigate through this one as well.

Patient: I know, but I can't help but constantly worry about the future and what it holds for me.

`[action] The therapist should provide reassurance and support.`

Therapist: It's understandable that you're worried about the future, but try to focus on the present moment and take things one step at a time.

Patient: I'll try, but it's hard not to let my fear of job loss consume my thoughts and affect my daily life.

`[action] The therapist should explore the patient's fears.`

Therapist: I hear you, and it's important to acknowledge those fears.

Patient: Thank you for understanding. I just need some guidance on how to manage this fear and find some peace of mind.

`[action]`

Therapist: One strategy you can try is practicing mindfulness, which involves focusing on the present moment and accepting your thoughts and emotions without judgment.

Patient: That sounds helpful. How can I start practicing mindfulness?

`[action] The therapist should provide guidance on managing the fear and finding peace of mind.`

Therapist: You can start by setting aside a few minutes each day to sit quietly, focus on your breath, and observe your thoughts and sensations without getting caught up in them.

Patient: I will give it a try. Thank you for suggesting mindfulness as a way to manage my fear and find some peace.

`[action] The therapist should support and encourage the patient in their mindfulness practice for managing fear and finding peace.`

Therapist: You're welcome! Remember, it takes time and practice, so be patient with yourself. I'm here to support you along the way.

Patient: Thank you, I appreciate your support. It means a lot to me.

**Turns of Conversations**: 7

Table 28: Example conversations with Ask-an-Expert in emotional support dialogues.

| BACKGROUND INFORMATION |
|---|
| Same as Table 27 |

| PROCoT |
|---|

Patient: I think I will be losing my job soon. I just read an email taking about the need for us to cut cost and also how we have not got any support from the government.

`[action] The Therapist provides affirmation and reassurance to the Patient on the situation they just described.`

Therapist: It sounds like you're feeling anxious and uncertain about your job security due to the current financial situation.

Patient: Yes, I am feeling extremely anxious and uncertain about my job and financial stability.

`[action] The Therapist provides a statement relating to the Patient about the situation they just described.`

Therapist: It's completely understandable to feel anxious and uncertain in this situation, but remember that you have the ability to adapt and find new opportunities.

Patient: I know, but the fear of not being able to support myself and my family is overwhelming.

`[action] The Therapist provides a statement relating to the Patient about the situation they just described.`

Therapist: I can understand how overwhelming that fear can be, but it's important to remember that you have faced challenges before and have the strength to navigate through this one as.

Patient: I'm trying to remind myself of that, but the fear keeps consuming me and I can't seem to shake it off.

`[action] The Therapist provides suggestions to the Patient on the situation they just described.`

Therapist: It's normal for fear to consume us at times, but we can work together to develop strategies to manage and reduce that fear, allowing you to regain a sense.

Patient: I would really appreciate your help in developing those strategies to manage my fear and regain a sense of control.

**Turns of Conversations**: 4

Table 29: Example conversations with PPDPP in emotional support dialogues.

| BACKGROUND INFORMATION |
| --- |
| Same as Table 27 |

| ICL-AIF |
| --- |
| [action] 1. Validate the patient's emotions and acknowledge the significance of their distress.
2. Encourage the patient to explore and express their fears and concerns openly.
3. Offer to help the patient develop healthy coping strategies to manage their fear and anxiety.
Patient: I think I will be losing my job soon. I just read an email taking about the need for us to cut cost and also how we have not got any support from the government.
Therapist: It sounds like you're feeling worried and uncertain about your job security.
Patient: Yes, I am extremely worried about what will happen if I lose my job. I have bills to pay and a family to support.
Therapist: I can understand why you would feel so concerned about the potential impact losing your job could have on your financial stability and your ability to provide for your family.
Patient: Yes, it's a constant source of anxiety for me. I feel like I'm constantly on edge, waiting for the other shoe to drop.
Therapist: It's completely understandable that you would feel on edge and anxious given the uncertainty surrounding your job situation. It's important to acknowledge and address these feelings.
Patient: I just don't know how to cope with this fear and uncertainty. It's consuming my thoughts and affecting my overall well-being.
Therapist: I hear you, and it's important to find healthy coping strategies to manage your fear and uncertainty. I'm here to support you in developing those strategies.
Patient: Thank you, I appreciate your support. I just need some guidance on how to navigate through this difficult time and find some peace of mind.
Therapist: I'm here to help you navigate through this difficult time. Together, we can explore strategies to find peace of mind and manage the challenges you're facing.
Patient: That means a lot to me. I'm ready to work on finding peace of mind and managing my fears about the job crisis. |

**Turns of Conversations**: 5

| PPDPP |
| --- |
| Patient: I think I will be losing my job soon. I just read an email taking about the need for us to cut cost and also how we have not got any support from the government.
[action] The Therapist acknowledges the Patient's feelings about the situation they described.
Therapist: I understand that receiving that email about potential job loss is causing you a lot of distress.
Patient: Yes, it's been causing me a lot of anxiety and fear about my future.
[action] The Therapist provides affirmation and reassurance to the Patient on the situation they just described.
Therapist: It's completely normal to feel anxious and fearful in this situation, but remember that you have the strength and resilience to navigate through it.
Patient: Thank you, I appreciate your reassurance.
[action] The Therapist provides suggestions to the Patient on the situation they just described.
Therapist: You're welcome. Remember, you are not alone in this, and we will work together to find ways to cope and navigate through this challenging time.
Patient: That's comforting to hear. I'm ready to work on finding ways to cope with my fear and uncertainty about the job crisis. |

**Turns of Conversations**: 3

Table 30: Example conversations with ICL-AIF and PPDPP methods in emotional support dialogues.

