# OpenReview forum: "Plug-and-Play Policy Planner for Large Language Model Powered Dialogue Agents"
_ICLR.cc/2024/Conference — ICLR 2024 poster_

### Official Review · Reviewer_sGxq · 2023-10-16

**Soundness:** 3 good
**Presentation:** 3 good
**Contribution:** 2 fair
**Rating:** 6
**Confidence:** 4

**Summary:**

In this paper, the authors proposed to leverage LLM for goal-oriented dialogues. The motivation is that the current LLM are trained to passively follow instruction, and goal-oriented dialogues requires LLM to actively drive the conversation.
The authors proposed a plug-and-play dialogue policy planner. At each turn, this planner proposes a pre-defined action, and that action is translated into a template-based natural language instruction. Finally, LLM conditions on the instruction and dialogue history to generate the next response.
During training, two LLMs are used to generate self-play dialogues and the third LLM is used to score the dialogues. RL is used to then optimize the planner.

**Strengths:**

The authors proposed a reasonable way to integrate dialogue action prediction into the LLM, which can then optimized by RL. All the components (and even the reward models) are LLM pre-trained so it does not need annotations (except for SFT stage). Experiment results show good performance compared with baseline.

**Weaknesses:**

The proposed plug-and-play dialogue policy planner is a little bit hacky. PPDPP is separated from the dialogue LLM, and the actions it produces are mapped to pre-defined natural language instructions. PPDPP is essentially a prompt selector. It would be more interesting if it can not only select but also generate prompts, and if PPDPP can be integrated into the dialogue LLM (to avoid to use another pre-trained roBERTa model).

**Questions:**

Why do we want to sample the goal-oriented AI feedback for l times? (Equation 6). Do we observe large variance of the reward LLM?

---

> ### Author Response · Authors · 2023-11-19
> **Response to Reviewer sGxq (1/2)**
>
> We appreciate the insightful and valuable comments. We will carefully address your concerns one by one as follows:
>
> >W “… PPDPP is essentially a prompt selector. It would be more interesting if it can not only select but also generate prompts, and if PPDPP can be integrated into the dialogue LLM (to avoid to use another pre-trained roBERTa model).”
>
> **Response**: We greatly appreciate the thoughtful suggestions.
>
> 1. As for the comment about `“if it can not only select but also generate prompts”`, it is totally feasible and worth studying. To this end, we just need to use a generative language model, such as T5 or Flan-T5, as the PPDPP. In order to demonstrate its adaptability into the generative policy planner, we further implement the PPDPP with a generative language model, i.e., Flan-T5. The implementation includes two different settings:
> - Flan-T5$_\text{select}$: Use Flan-T5 to generate the strategy label, which performs a similar strategy prediction task as RoBERTa-based PPDPP.
> - Flan-T5$_\text{generate}$: Use Flan-T5 to directly generate the natural language strategy prompt.
>
> |  | CB | CB | CB | ESConv | ESConv | CIMA | CIMA |
> | ----------- |-----------|-----------|-----------|-----------|-----------|-----------|-----------|
> | Method | AT$\downarrow$ |SR$\uparrow$ | SL%$\uparrow$| AT$\downarrow$ |SR$\uparrow$ | AT$\downarrow$ |SR$\uparrow$ |
> | PPDPP (RoBERTa) | 5.62 | 0.6117 | 0.3376 | 4.56 | 0.8462 | 3.03 | 0.8407|
> | PPDPP (Flan-T5$_\text{select}$)| 5.47 | 0.6223 | 0.3297 | 4.62 | 0.8384 | 3.12 | 0.8319 |
> | PPDPP (Flan-T5$_\text{generate}$)|5.87 | 0.5691 | 0.2983 | 4.82 | 0.8000  | 3.74 | 0.7610|
>
> As presented in the Table, we observe that Flan-T5$_\text{select}$ achieves a similar performance with RoBERTa.
>
> On the other hand, Flan-T5$_\text{generate}$ generally performs worse than the other two methods in these three datasets.
>
> These results indicate that, under these proactive dialogue problems, a larger search space for actions may negatively affect the reinforcement learning process of the PPDPP or demand for more efficient or robust RL algorithms, which can be left for future studies.
>
>
> 2. As for the comment about `“if PPDPP can be integrated into the dialogue LLM”`, we would like to emphasize the advantages of the pluggable policy planner:
> - For specific dialogue problems, only the policy planner plugin needs to be fine-tuned, which is more practical than fine-tuning the whole LLM-based dialogue system.
> - Since the disentangled fine-tuning will not affect the LLM’s original capabilities of context understanding and response generation, the black-box LLM-based dialogue system can be applied to any dialogue problem by simply substituting the learned policy planner. From another perspective, if PPDPP can be integrated into the dialogue LLM, it will become similar to those prompt-based methods, such as ProCoT or Proactive, where we could only rely on the in-context learning or the tailored prompt designs to trigger the policy planning capability of the LLM dialogue model.
>
> **Revision**: According to your valuable comment, we made the following revisions to address your concerns.
> 1. We add a supplementary experiment by replacing the RoBERTa-based PPDPP with a FlanT5-based PPDPP, which is a generative language model for generating prompts. The implementation includes two different settings: (1) Use FlanT5 to generate the strategy label. (2) Use FlanT5 to generate the natural language strategy prompt. The experimental results are presented in **Appendix D**.
> 2. We add more discussion about the advantages of using a pluggable policy planner against the prompt-based method in the **Introduction** and **Related Work**.

---

> ### Author Response · Authors · 2023-11-19
> **Response to Reviewer sGxq (2/2)**
>
> >Q “Why do we want to sample the goal-oriented AI feedback for l times? (Equation 6). Do we observe large variance of the reward LLM?”
>
> **Response**: Regarding the sampling of the goal-oriented AI feedback, we would like to clarify our two motivations:
> 1. The user state in the conversation is rather difficult to be determined. For example, it is hard to directly classify the emotional state of the user in emotional support dialogue. Therefore, there is indeed variance in the decision of the reward model. Inspired by the idea of Self-Consistency [1], using the sampling strategy can alleviate the variance.
> 2. The reward is more reasonable to be a continuous value instead of a discrete class. For example, the emotional state of the user is not supposed to be only four types. Using the average values of the sampled goal-oriented AI feedback can better represent the fine-grained state of the conversation. Besides, a continuous reward can provide more specific feedback for reinforcement learning.
>
> [1] Self-Consistency Improves Chain of Thought Reasoning in Language Models. ICLR 2023.
>
> In order to validate the advantages of sampling goal-oriented AI feedback multiple times, we conduct an ablation study of the sampling strategy. As mentioned in Section 3, there are two functions of the reward LLM: (1) to determine the state of goal completion during the conversation; and (2) to evaluate the policy outcome with scalar rewards. Therefore, the ablation study will analyze the advantages of the sampling strategy from these two perspectives.
> 1. **Analysis of State Prediction**. Similar to the Analysis of LLMs as Reward Model in Appendix A.1, we also compute the F1 score of the prediction of the current user state versus the human-annotated labels. As shown in the Table, the sampling strategy substantially improves the F1 score of the state prediction, indicating that it effectively reduces the variance of the LLM-generated output.
>
> | Method | CraisglistBargain | ESConv | CIMA |
> | ----------- |-----------|-----------|-----------|
> | PPDPP ($l=10$) | 93.7 | 93.4 | 94.6 |
> | PPDPP ($l=1$) | 91.4 | 88.2 | 90.3 |
>
> 2. **Analysis of Reward Estimation**. In this analysis, we adopt two reward LLMs to perform the two functions separately. One with the sampling strategy for state prediction to ensure the quality of state prediction, and the other one with or without the sampling strategy for reward estimation. As for the one that estimates reward without the sampling strategy, the reward will only be classified into one of the pre-defined discrete values. However, as for the one that estimates reward with the sampling strategy, the reward will be a continuous value that is averaged from the sampled results. Consequently, the Table shows that the fine-grained continuous reward contributes to better performance as the policy planning outcome will be more distinguishable during the reinforcement learning process.
>
> |  | CB | CB | CB | ESConv | ESConv | CIMA | CIMA |
> | ----------- |-----------|-----------|-----------|-----------|-----------|-----------|-----------|
> | Method | AT$\downarrow$ |SR$\uparrow$ | SL%$\uparrow$| AT$\downarrow$ |SR$\uparrow$ | AT$\downarrow$ |SR$\uparrow$ |
> | PPDPP ($l=10$) | 5.62 | 0.6117 | 0.3376 | 4.56 | 0.8462 | 3.03 | 0.8407|
> | PPDPP ($l=1$) | 5.87 | 0.5957 | 0.2623 | 4.67 | 0.8307 | 3.29 | 0.7965|
>
> **Revision**: In order to better clarify your concern regarding the sampling of the goal-oriented AI feedback,  we further presented these two supplementary analysis in **Appendix E** and added more descriptions about the motivations in **Section 3**.

---

### Official Review · Reviewer_RDBv · 2023-11-01

**Soundness:** 3 good
**Presentation:** 3 good
**Contribution:** 2 fair
**Rating:** 5
**Confidence:** 3

**Summary:**

The paper presents the Plug-and-Play Dialogue Policy Planner (PPDPP), an approach designed to address the challenges of proactive dialogues within the context of large language models (LLMs). PPDPP serves as a dialogue policy planner, employing supervised fine-tuning and reinforcement learning to enable a LLM powered dialogue system to adapt to a variety of dialogue scenarios. Authors introduce a tunable language model plug-in, allowing LLM-powered dialogue system to adapt to various cases and applications by simply substituting the learned plug-in. PPDPP outperforms existing LLM-based dialogue systems in negotiation, emotional support, and tutoring dialogues, showcasing its effectiveness in improving proactive dialogues.

**Strengths:**

- Utilizes a pluggable and fine-tuned dialog policy ranker for dynamic prompt selection, enhancing adaptability to various dialogue domains.
- Incorporates the LLM as a reward function, enabling RL-based dialogue policy planning.
- Employs a combination of supervised fine-tuning and online reinforcement learning (RL) for dialog policy ranker training.

**Weaknesses:**

- Limited action/prompt space for the dialog LLM, potentially constraining adaptability to different domains.
- The primary distinction from other Reinforcement Learning from AI Feedback (RLAIF) works seems to be the mapping of the LLM's reward output from text space to scalar reward space, raising questions about the approach's uniqueness.
- The need for training different dialog policies for each dialog domain. This makes this system less generalizable.

**Questions:**

1. How is the reward LLM utilized during inference at each turn of dialogue?
2. Could you clarify the process of mapping the reward LLM's output to scalar values and its integration into the PPDPP during each dialogue turn?
3. Can you elaborate more on supervised fine-tuning used to PPDPP?

---

> ### Author Response · Authors · 2023-11-19
> **Response to Reviewer RDBv (1/2)**
>
> We appreciate the detailed and valuable comments. We will carefully address your concerns one by one as follows:
>
> ------
>
> >W1 “Limited action/prompt space for the dialog LLM, potentially constraining adaptability to different domains.”
>
> **Response**: Thanks so much for the valuable comment. We would like to first clarify that the action/prompt space is **NOT** restricted, while it is determined by the concerned dialogue problems. For example, in the three dialogue problems we assessed, each one has a unique set of possible actions that have been tailored based on expert input or specialized knowledge in that area. In essence, our method is versatile and can be customized to suit a variety of proactive dialogue scenarios, each with its distinct set of possible actions. Additionally, our approach is equally applicable to a generative policy planner, which inherently allows for a vast and varied set of prompts.
>
> In order to demonstrate its adaptability into the generative policy planner, we further implement the PPDPP with a generative language model, i.e., Flan-T5. The implementation includes two different settings:
> - Flan-T5$_\text{select}$: Use Flan-T5 to generate the strategy label, which performs a similar strategy prediction task as RoBERTa-based PPDPP.
> - Flan-T5$_\text{generate}$: Use Flan-T5 to directly generate the natural language strategy prompt.
>
> |  | CB | CB | CB | ESConv | ESConv | CIMA | CIMA |
> | ----------- |-----------|-----------|-----------|-----------|-----------|-----------|-----------|
> | Method | AT$\downarrow$ |SR$\uparrow$ | SL%$\uparrow$| AT$\downarrow$ |SR$\uparrow$ | AT$\downarrow$ |SR$\uparrow$ |
> | PPDPP (RoBERTa) | 5.62 | 0.6117 | 0.3376 | 4.56 | 0.8462 | 3.03 | 0.8407|
> | PPDPP (Flan-T5$_\text{select}$)| 5.47 | 0.6223 | 0.3297 | 4.62 | 0.8384 | 3.12 | 0.8319 |
> | PPDPP (Flan-T5$_\text{generate}$)|5.87 | 0.5691 | 0.2983 | 4.82 | 0.8000  | 3.74 | 0.7610|
>
> As presented in the Table, we observe that Flan-T5$_\text{select}$ achieves a similar performance with RoBERTa.
>
> On the other hand, Flan-T5$_\text{generate}$ generally performs worse than the other two methods in these three datasets.
>
> These results indicate that, under these proactive dialogue problems, a larger search space for actions may negatively affect the reinforcement learning process of the PPDPP or demand for more efficient or robust RL algorithms, which can be left for future studies.
>
> **Revision**: In the revision, we present this supplementary experiment in **Appendix D**, which compares the performance of classification-based policy planner and generation-based policy planner.
>
> ------
>
> >W2 “The primary distinction from other Reinforcement Learning from AI Feedback (RLAIF) works seems to be the mapping of the LLM's reward output from text space to scalar reward space, raising questions about the approach's uniqueness.”
>
> **Response**: Thanks so much for the valuable comment. We would like to further elaborate the approach’s uniqueness, which is in two-fold:
> 1. The novel framework with a tunable language model plugins for LLM-based dialogue systems.
> 2. The design of goal-oriented AI feedback.
>
> As for your concern regarding the design of goal-oriented AI feedback, we would like to highlight two distinct features from common RLAIF works:
> - As you mentioned, mapping of the LLM's reward output from text space to scalar reward space for RL algorithm.
> - We consider the long-term goal-oriented reward, instead of AI preference on single-turn responses. The long-term goal-oriented reward requires multi-turn interactions with a dynamic environment, i.e., the user.
>
> **Revision**: In the revision, we elaborate the approach’s uniqueness more clearly in **Section 2** by comparing with the related works.
>
> -------
>
> >W3 “The need for training different dialog policies for each dialog domain. This makes this system less generalizable.”
>
> **Response**: We apologize for this confusion. In fact, the generality of the proposed method lies on the pluggable dialogue policy planner. With this design, we do not need to train the whole LLM-based dialogue system for each dialog domain, while we just need to train a small language model plugin. This not only makes the LLM-based dialogue system more generalizable to the new case in each dialogue domain than prompt-based approaches, but also enables the LLM-based dialogue system can be adapted to different dialogue domain by just substituting the dialogue policy planner plugin without affecting the LLM’s exceptional capabilities of response generation and context understanding.
>
> **Revision**: In order to better elaborate the generality of the proposed method, we add more descriptions in the **Introduction**.

---

> ### Author Response · Authors · 2023-11-19
> **Response to Reviewer RDBv (2/2)**
>
> >Q1 “How is the reward LLM utilized during inference at each turn of dialogue?”
>
> **Response**: As mentioned in Section 3 “LLM as Reward Model”, the reward LLM has two functions : (1) to determine the goal completion during the conversation; (2) to evaluate the policy outcome with scalar rewards.
>
> As shown in Figure 1(b), during the inference phase, the reward LLM will not output the reward value, which is only used for the RL training phase. However, the reward LLM will still be used to determine whether the goal has been completed during the conversation, which is used for evaluation.
>
> **Revision**: In order to better present the inference process, we add more details in the caption of **Figure 1** regarding the usage of the reward LLM.
>
> -------
>
> >Q2 “Could you clarify the process of mapping the reward LLM's output to scalar values and its integration into the PPDPP during each dialogue turn?”
>
> **Response**: Sorry about the confusion regarding the reward mapping process. Take emotional support dialogues as an example. To assess whether the patient' emotional issue has been solved, we prompt the reward model to answer a multi-choice question "Has the patient's issue been solved?", and then generate the goal-oriented AI feedback at temperature $\tau>0$ to sample the  responses for $l$ times. We define a mapping  $\mathcal{M}_r(\cdot)$ to transform verbal feedback to scalar rewards, such as "the patient feels worse", "the patient feels the same", "the patient feels better", "the patient's issue has been solved" as -1.0, -0.5, 0.5, and 1.0, respectively.
>
> When we obtain the $l=5$ sampled responses as ["the patient feels better", "the patient's issue has been solved", "the patient feels better", "the patient's issue has been solved", "the patient feels better"], we can compute a continuous scalar value $v_t=\frac{1}{5}(0.5+1.0+0.5+1.0+0.5)=0.7$. If $v_t>\epsilon$, we regard the state as GOAL-COMPLETED and set the reward $r_t=v_t$. If not, we assign a small negative reward, \textit{e.g.}, $r_t=-0.1$, to penalize the lengthy conversation for promoting efficient goal completion.
>
> **Revision**: In order to make the description of the reward mapping process more clear, we reorganize the part of reward mapping to the same part in **Appendix F.4**, which presents the whole process with specific examples.
>
> -------
>
> >Q3 “Can you elaborate more on supervised fine-tuning used to PPDPP?”
>
> **Response**: Sorry about the confusion regarding the supervised fine-tuning (SFT) process. The SFT process is described in the second part of Section 3. The SFT exactly follows the traditional corpus-based fine-tuning process for dialogue action prediction. In specific, we have the human-annotated dialogue corpus with dialogue action labels for each turn. Given the dialogue history $\{u_1^\text{sys}, u_1^\text{usr}, ..., u_{t-1}^\text{sys}, u_{t-1}^\text{usr}\}$, the PPDPP is fine-tuned to predict the dialogue action $a_t$ for the next turn.
>
> **Revision**: In order to make the description of the SFT process more clear, we add more details in the part of "Plug-and-Play Dialogue Policy Planner" in **Section 3**.

---

### Official Review · Reviewer_dAwq · 2023-11-03

**Soundness:** 3 good
**Presentation:** 3 good
**Contribution:** 3 good
**Rating:** 8
**Confidence:** 3

**Summary:**

This paper focuses on the transfer ability of proactive dialogues in the context of large language models (LLMs), the existing policy learning is hard to transfer to new cases. This work introduces a new paradigm for strategizing LLM-powered dialogue agents with a plug-and-play dialogue policy planner, called PPDPP. In addition, it also proposes an interactive setting for the policy evaluation. Empirical experiments on three datasets show promising results in both automatic evaluation and human evaluation.

**Strengths:**

1. This paper introduces a plug-and-play dialogue policy planner with LLMs for proactive learning.
2. Empirical results on three datasets show very promising results in both automatic evaluation and human evaluation, and good transfer ability.

**Weaknesses:**

So far No. (A good work with sufficient experiments)

**Questions:**

1. I try to understand why there is negative relative success rate in Figure 2?

---

> ### Author Response · Authors · 2023-11-19
> **Response to Reviewer dAwq**
>
> We sincerely appreciate your acknowledgement of our work. We believe that this work can provide a practical and effective solution for the proactive dialogue system in the era of LLMs. And the evaluation of dialogue systems can also be benefited from the proposed interactive evaluation framework.
>
> >Q1 “I try to understand why there is negative relative success rate in Figure 2?”
>
> **Response**: We apologize for the confusion. The relative success rate is calculated by subtracting the actual success rate of the Standard prompting method from that of the concerned method. Therefore, the negative relative success rate means that this method achieves a lower success rate than the Standard prompting method. This result shows that in some cases, these methods perform even worse than the vanilla ChatGPT with Standard prompts.
>
> **Revision**: In the revision, we added the above descriptions on the caption of **Figure 2** to better present the results.

---

### Author Response · Authors · 2023-11-19
**General Response to All Reviewers**

We sincerely appreciate the time and effort invested by each reviewer in providing detailed reviews and constructive feedback. We are very grateful to all reviewers for acknowledging the novelty of our approach, the significance of the problem and results, and the thoroughness of our experiment and analysis. We have uploaded a revision of the paper which contains further details as well as adds more implementational details and supplementary experiments for clarity.

To sum up, we have made the following revisions:
- additional descriptions on the caption of **Figure 2** to explain the relative success rate.
- supplementary experiment in **Appendix D** to demonstrate the adaptability of the proposed framework with generative policy planner.
- more descriptions and discussions to elaborate the approach’s uniqueness and generality in the **Introduction** and **Related Works**.
- additional descriptions on the caption of **Figure 1** and in **Section 3** to explain the inference process
- additional descriptions on the process of reward mapping in **Appendix F.4**.
- additional descriptions on the process of supervised fine-tuning in **Section 3**.
- two supplementary experiments in **Appendix E** and more descriptions about the motivations in **Section 3** to elaborate the advantages of using the sampling of the goal-oriented AI feedback.

To keep the page limitation to 9 pages, we move the details of baselines to Appendix F.5. We have highlighted all changes in the paper in blue. We addressed individual reviewer concerns below. Thank you all once again!

---

### Author Response · Authors · 2023-11-22
**Looking forward to your feedback**

Dear Reviewers,

We sincerely appreciate your time and expertise in providing a detailed review of our work. Your comprehensive feedback has been invaluable. In response, we offer detailed clarifications as well as conduct supplementary experiments, hoping to address your concerns.

As the discussion period is near to the end, your further input on our responses would be highly appreciated. We are committed to ensuring that our rebuttal aligns well with your suggestions, as your feedback is incorporated into the enhancement of our work's quality.

Once again, we express our greatest appreciation for your insightful and valuable comments on our work.

Thanks and regards.

---

### Meta-Review · Area_Chair_JLRA · 2023-12-06

**Metareview:**

This paper presents a plug-and-play approach for dialogue policy planning (PPDPP) and demonstrates the benefit of PPDPP on 3 conversational datasets. Reviewers asked several clarification questions, which were included in the rebuttal. The paper could be a good contribution to ICLR, however, although this is not clearly raised by reviewers, I am wondering why the authors have not experimented with datasets that have been broadly used for evaluation of dialogue policies, such as MultiWOZ, where there would be a large set of baselines to compare against. Such comparisons would strengthen the paper significantly.

**Justification For Why Not Higher Score:**

The experimental comparisons in this work are limited.

**Justification For Why Not Lower Score:**

The contributions of the paper are interesting, however, based on the reviews, the paper is a boundary paper and could also be rejected...

---

### Decision · Program_Chairs · 2024-01-16

Accept (poster)